# Get Rid of Isolation: A Continuous Multi-task Spatio-Temporal Learning Framework

**Zhongchao Yi[1], Zhengyang Zhou[1,2,3,*], Qihe Huang[1], Yanjiang Chen[1],**
**Liheng Yu[1], Xu Wang[1,2], Yang Wang[1,2,*]**
[1]University of Science and Technology of China (USTC), Hefei, China
[2]Suzhou Institute for Advanced Research, USTC, Suzhou, China
[3]State Key Laboratory of Resources and Environmental Information System, Beijing, China
`{zhongchaoyi, hqh, yjchen, yuliheng}@mail.ustc.edu.cn,`
`{wx309, zzy0929*, angyan*}@ustc.edu.cn`

## Abstract

Spatiotemporal learning has become a pivotal technique to enable urban intelligence. Traditional spatiotemporal models mostly focus on a specific task by assuming a same distribution between training and testing sets. However, given that urban systems are usually dynamic, multi-sourced with imbalanced data distributions, current specific task-specific models fail to generalize to new urban conditions and adapt to new domains without explicitly modeling interdependencies across various dimensions and types of urban data. To this end, we argue that there is an essential to propose a Continuous Multi-task Spatio-Temporal learning framework (CMuST) to empower collective urban intelligence, which reforms the urban spatiotemporal learning from single-domain to cooperatively multi-dimensional and multi-task learning. Specifically, CMuST proposes a new multi-dimensional spatiotemporal interaction network (MSTI) to allow cross-interactions between context and main observations as well as self-interactions within spatial and temporal aspects to be exposed, which is also the core for capturing task-level commonality and personalization. To ensure continuous task learning, a novel Rolling Adaptation training scheme (RoAda) is devised, which not only preserves task uniqueness by constructing data summarization-driven task prompts, but also harnesses correlated patterns among tasks by iterative model behavior modeling. We further establish a benchmark of three cities for multi-task spatiotemporal learning, and empirically demonstrate the superiority of CMuST via extensive evaluations on these datasets. The impressive improvements on both few-shot streaming data and new domain tasks against existing SOAT methods are achieved. Code is available at `https://github.com/DILab-USTCSZ/CMuST`.

## 1 Introduction

Spatiotemporal learning has become a pivotal technique to enable smart and convenient urban lives, benefiting diverse urban applications from intra-city travelling, environment controlling to location-based POI recommendation, and injecting the vitality into urban economics. Existing spatiotemporal learning solutions [19, 9, 38, 40, 51, 42, 37, 10, 8] focus on improving performances of a task-specific model independently where these methods devise various spatial learning blocks [44, 8, 43] and temporal dependency extraction modules [40, 34, 9] to model the spatiotemporal heterogeneity.

Actually, urban spatiotemporal systems are usually highly dynamic with emerging new data modality, leading to serious generalization issue on both data pattern and task adaptation. As illustrated in

---

*Yang Wang and Zhengyang Zhou are corresponding authors.

38th Conference on Neural Information Processing Systems (NeurIPS 2024).

Figure 1, the traffic volume patterns can evolve with urban expansion and establishment of new POIs. Concurrently, with increasing attention on road safety, traffic accident prediction has become a new task in intelligent transportation that inevitably suffers from the cold-start issue. Unfortunately, traditional task-specific models usually assume that data on a single task follows independent and identical distribution and are intensively available where such assumption directly leads to failures on data sparsity scenarios and generalization to new tasks. In fact, given diverse datasets, separately training single task-specific spatiotemporal models is cost sensitive and will trap the models into isolation. To this end, we argue that a continuous multi-task spatiotemporal learning framework is highly desirable to facilitate the task-level cooperation. It is even more interesting and exciting to jointly model multi-domain datasets with multi-task learning, which empowers understanding spatiotemporal system in a holistic perspective and reinforce each individual task by exploiting the collective intelligence from diverse data domains.

The key towards exploiting task-wise correlations for mutual improvement is to capture the common interdependencies across data dimensions and domains. Current multi-task learning schemes either investigate the regularization effects between auxiliary and main tasks [31, 17], or devise loss objectives to constrain the consistency between each task [7, 32].

Actually, given a spatiotemporal domain, there must be common interdependencies across different data types and domains, which are valuable for cooperated learning. Even prosperity of multi-task learning and spatiotemporal forecasting, there are never a systematic solution on how various sourced data from different tasks reinforce a specific task with multi-task learning. More specifically, interrelations among obser-

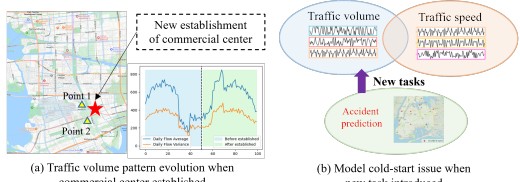

(a) Traffic volume pattern evolution when commercial center established   (b) Model cold-start issue when new task introduced

Figure 1: Illustration of evolution on data patterns and learning tasks.

vations in a system can be decomposed into multi-dimensional interactions, i.e., from contextual environment to respective spatial relations and temporal evolutions, spatial-temporal level interactions and relations across different data domains. Considering the multi-level semantic correlations, learning the system in a holistic perspective is required and further pose two challenges to a continuous multi-task spatiotemporal learning framework, i.e., 1) How to disentangle complex associations between data dimensions and domains, and capture such dependencies in an adaptive manner to improve spatiotemporal representation, hence facilitating the extraction of common patterns for mutual enhancement. 2) How to exploit task-level commonality and personality to jointly model the multi-task datasets, and exploit such extracted task-level commonality and diversity to reinforce respective task for getting rid of task isolation.

In our work, a Continuous Multi-task SpatioTemporal learning framework, CMuST, is proposed to jointly model multiple datasets in an integrated urban system thus reinforcing respective learning tasks. Specifically, a Multi-dimensional Spatio-Temporal Interaction Network (MSTI) is first devised to dissect interactions across data dimensions, including context-spatial, context-temporal and self-interaction within spatial and temporal dimensions. MSTI enables improved spatiotemporal representation with interactions, and also provides disentangled patterns to support commonality extraction. After that, a Rolling Adaptation training scheme, RoAda, which iteratively captures the task-wise consistency and task-specific diversity, is proposed. In RoAda, to maintain task characterization, a task-specific prompt is constructed to preserve unique patterns distinguishing from other tasks by compressing data patterns via an AutoEncoder. To capture the commonality across tasks, we propose a weight behavior modeling strategy to iteratively highlight the minimized variations of learnable weights, i.e., stable interactions during continuous training, which encapsulates crucial task-level commonalities. This approach not only stabilizes learning through continuous task rolling, but alleviates the cold-start over new tasks with shared patterns. Finally, a task-specific refinement is devised to leverage commonality and fine-grained adaptation on specific tasks.

The contributions of this work can be three-fold. 1) The first continuous multi-task spatiotemporal learning framework, CMuST to jointly model learning tasks in a same spatiotemporal domain, which not only reinforces individual correlated learning task in collective perspective, but also help understand the cooperative mechanism of dynamic spatiotemporal systems. 2) Technically, two learning modules, MSTI and RoAda are proposed to dissect the impacts and interactions over multi-dimensions, and iteratively update the task-wise commonality and generate individual personalization to continuous task adaptation in multi-task learning. 3) We construct benchmark datasets in each of

three cities, where two of them consists of at least 3 types of observations within same spatiotemporal domain. The extensive experiments demonstrate the superiority on enhancement of each individual task with limited data and the interpretation of task-wise continuous learning.

## 2    Related Work

**Spatiotemporal forecasting** is an emerging technique to capture the dynamic spatial and temporal evolution for diverse urban predictions, where the methods can be divided into machine learning-based and deep learning-based. Conventional solutions rely on complex mathematical tools to simulate the dynamics including ARIMA [25], SVR [3] and matrix-factorization learning [28] for capturing spatial correlations. With deep learning solutions flourishing, Convolution Neural Networks (CNNs) [40, 11, 46] are exploited to imitate the temporal dependencies and GNNs [43, 13, 36] are utilized to imitate spatial propagation. Meanwhile, by taking the advantage of the flexibility and interpretability of attention, Spatial-Temporal Attention [9], and Vision Transformer (ViT) [6] are introduced to improve spatiotemporal representation. Also there are also many methods for temporal periodicity capturing [12, 20]. More specifically, DG2RNN [49] designs a dual-graph convolutional module to capture local spatial dependencies from both road distance and adaptive correlation perspectives. PDFormer [15] designs a spatial self-attention and introduces two graph masking matrices to highlight the spatial dependencies of short- and long-range views. TESTAM [18] uses time-enhanced ST attention by mixture-of-experts and modeling both static and dynamic graphs. Even so, most solutions focus on single-task intelligence, fail to deal with complex interactions between data dimensions and never extract task-level commonality patterns, resulting in inferior performances on exploiting collective intelligence over multiple tasks. In contrast, we merge the gap by disentangling learnable interaction patterns and exploring rolling task adaptations.

**Multi-task learning.** Plenty of efforts have been made on multi-task learning (MTL) and MTL can be elaborated by two-fold, i.e., feature-based and parameter-based. Feature-based MTL [2, 24] learns a common feature representation for different tasks, but it may be easily affected by outlier tasks. To this end, parameter-based MTL [30, 16] is devised to exploit model parameters to relate different tasks, which is expected to learn robust parameters. Majority of these MTL schemes either concentrate on the diversity design and regularization effects of auxiliary tasks to main task [30, 27], or construct loss functions to ensure task-wise consistency [47]. A pioneering work investigates a gradient-driven task grouping to realize multi-task learning [41] where the focuses are text and images using pre-trained CLIP model. For spatiotemporal learning, RiskOracle [52] and RiskSeq [53] are proposed to simultaneously predict multi-grained risks and auxiliary traffic elements. More recently, UniST [45] and UniTime [22] construct a unified model for spatiotemporal and time-series prediction. However, all researches on ST learning still never dissect task-wise correlations, especially capture explicit consistency and diversity among tasks and investigate how each task reinforce the core task, which is of great significance for performance and interpretability in MTL.

**Continuous learning and task continuous learning.** Continuous learning (CL) usually keeps long-term and important information while updates model memories with newly arrived instances [29, 5, 35]. For spatiotemporal forecasting, Chen, et, al. [5] proposes a historical-data replay strategy, TrafficStream, to update the neural network with all nodes feeding, while PECPM [35] manages a pattern bank with conflict nodes, which reduces the memory storage burdens. Most existing CL solutions are designed for homogeneous sourced data. Then there are very few research investigating task-level continuous learning where task can be converted from one to another. A pioneering work, CLS-ER [1] realizes class-level and domain-level continuous learning with a dual memory to respectively store instances and build long-short term memories. Even so, the data input to CLS-ER is with same image-like property and scales, which is still far away from task diversity. We argue that learning continuously on task levels can increase the data tolerance over low-quality data and also generalization capacity on new tasks. In this work, by taking bonus of continuous learning, we propose to iteratively assimilate the commonality and extract personalization to reinforce each learning task and realize continuous multi-task spatiotemporal intelligence.

## 3    Preliminary

***Definition 1 (Spatiotemporal features.)***    Spatiotemporal features refer to the data points collected by sensors deployed in urban environments, such as traffic dynamics on roads. We define a spatiotemporal

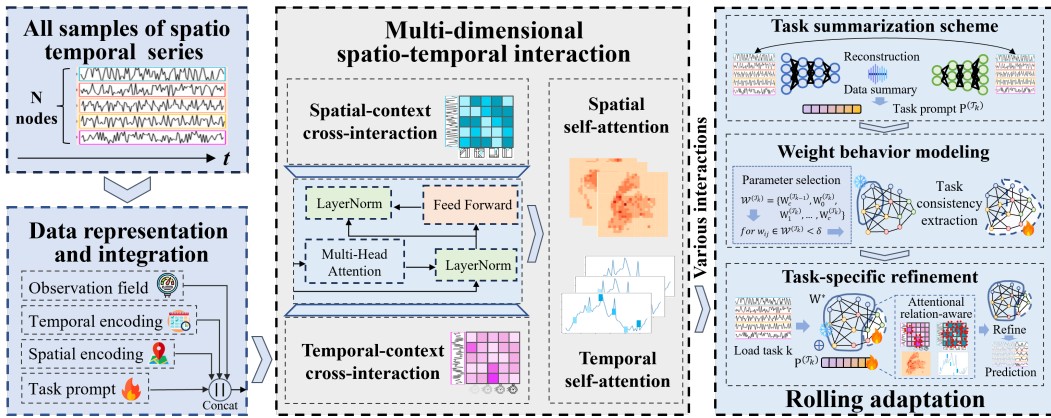

Figure 2: Framework overview of CMuST.

feature at a specific time and location as a vector $X \in \mathbb{R}^C$, where $C$ represents the number of attributes (e.g., vehicle flow, speed) recorded by the sensor. We generalize this to define spatiotemporal data collected over a period as $\mathbf{X} \in \mathbb{R}^{T \times N \times C}$, where $T$ denotes the number of discrete time intervals, and $N$ denotes the number of spatial nodes corresponding to sensor locations.

***Definition 2 (Spatiotemporal prediction.)*** Spatiotemporal prediction involves forecasting future values of spatiotemporal data based on historical observations. Specifically, given a historical dataset $\mathbf{X} = [X_{t-T}, \ldots, X_{t-1}, X_t] \in \mathbb{R}^{T \times N \times C}$, the objective is to predict future observations $\hat{\mathbf{Y}} = [X_{t+1}, \ldots, X_{t+T'}] \in \mathbb{R}^{T' \times N \times C}$. Here, $T'$ represents the prediction horizon, the number of future time steps we seek to forecast, which for simplicity is set equal to $T$ in this paper.

***Definition 3 (Multi-task spatiotemporal learning.)*** Given a set of spatiotemporal tasks in an integrated urban system $\mathbb{T} = \{\mathcal{T}_1, \mathcal{T}_2, \ldots, \mathcal{T}_k\}$ where each task $\mathcal{T}_i$ is associated with a dataset $\mathcal{D}_i = \{\mathbf{X}_i, \mathbf{Y}_i\}$. Here, $\mathbf{X}_i \in \mathbb{R}^{T_i \times N_i \times C_i}$ represents the input features collected over $T_i$ time steps, across $N_i$ spatial locations, and $C_i$ feature dimensions, and $\mathbf{Y}_i \in \mathbb{R}^{T'_i \times N_i \times C_i}$ represents the corresponding targets. Multi-task spatiotemporal learning aims to learn a function $\hat{f} : \mathbf{X}_0, \mathbf{X}_1, \ldots, \mathbf{X}_{k-1} \to \hat{\mathbf{Y}}_0, \hat{\mathbf{Y}}_1, \ldots, \hat{\mathbf{Y}}_{k-1}$ that optimally predicts all target $\mathbf{Y}_i$ from their respective inputs $\mathbf{X}_i$, leveraging shared and unique patterns across tasks to improve generalization performance.

## 4 Methodology

### 4.1 Framework Overview

The CMuST is crafted to advance urban intelligence through a synergistic integration of three components in Figure 2. We processes and standardizes diverse urban data into a harmonized format, and propose the MSTI to intricately dissect the complex interactions within spatiotemporal data, and devise a RoAda to dynamically fine-tune the model via continuous and careful updating, ensuring robust adaptability and consistent performance across fluctuating urban environments.

### 4.2 Data Representation and Integration

To harness various data domains within urban spatiotemporal systems as well as data interactions between diverse dimensions, the first task is to appropriately process main ST observations, spatial indicator, and temporal indicator to create a comprehensive and integrated data representation tailored for multi-task ST learning, enabling further interactive modeling between them.

To be specific, the main observation data, namely target of interest of urban datasets, are denoted as $\mathbf{X}_{obs} \in \mathbb{R}^{T \times N \times C}$, and then mapped into a spatiotemporal representation $\mathbf{E}_{obs} \in \mathbb{R}^{T \times N \times d_{obs}}$ via an MLP $\mathrm{ObsMLP}(\mathbf{X}_{obs}; \theta_{obs})$. Similarly, the spatial indicator $\mathbf{X}_s \in \mathbb{R}^{T \times N \times 2}$, consisting of longitude and latitude coordinates, is applied with a linear layer $\mathrm{SpatialMLP}(\mathbf{X}_s; \theta_s)$ to produce the spatial representation $\mathbf{E}_s \in \mathbb{R}^{T \times N \times d_s}$. Temporal indicator comprises day-of-week, time-of-day $\mathbf{X}_{dow}, \mathbf{X}_{tod} \in \mathbb{R}^{T \times N}$ and timestamps $\mathbf{X}_{ts} \in \mathbb{R}^{T \times N \times 6}$, which are further compressed into hidden

representation, i.e., $\mathbf{E}_t = \text{TemporalMLP}(\mathbf{E}_{ts}\|\mathbf{E}_{dow}\|\mathbf{E}_{tod}; \theta_t) \in \mathbb{R}^{T \times N \times d_t}$, enabling the model to capture the periodicity and sequential features of temporal data. $\mathbf{P}_\tau$ is a task-specific prompt [2] to ensure task-awareness, and is integrated into the final embedding. Given the task $\mathcal{T}_k$,

$$\mathbf{H}^{(\mathcal{T}_k)} = \mathbf{E}_{obs}\|\mathbf{E}_s\|\mathbf{E}_t\|\mathbf{P}^{(\mathcal{T}_k)} \tag{1}$$

where $\|$ denotes vector concatenation, combining spatial, temporal, and observational embeddings with the task prompt into a comprehensive representation $\mathbf{H}^{(\mathcal{T}_k)} \in \mathbb{R}^{T \times N \times d_h}$. We will use $\mathbf{H}$ as representation for a specific task in following sections.

### 4.3 Multi-dimensional Spatio-Temporal Interaction

Spatiotemporal observations are usually complex with multiple-level interactions where such interactions and correlations play vital roles in enhancing commonality learning through different learning domains. To this end, we devise an MSTI, to intricately dissect and disentangle interactions within spatiotemporal data from spatial-temporal indicators to main observations by inheriting nice property of attention mechanisms, which all utilize transformed slices from the integrated representation $\mathbf{H}$.

**Spatial-context cross-interaction.** To quantitatively investigate how spatial indicator interact with main observations, we devise a multi-head cross-attention [14] architecture (MHCA) where spatial and observational components are alternately used as queries ($\mathbf{Q}$) and key-value ($\mathbf{KV}$) pairs:

$$\text{MHCA}^{(a,b)}(\mathbf{X}) = \bigg\|_{h=1}^{\text{head}} \text{CrossAttention}^{(a,b)}(h)\mathbf{W}^{\mathbf{O}} \tag{2}$$

$$\text{CrossAttention}^{(a,b)}(h) = \text{Attention}\left(\mathbf{Q}_h^{(a)}, \mathbf{K}_h^{(b)}, \mathbf{V}_h^{(b)}\right) \tag{3}$$

$$\text{Attention}\left(\mathbf{Q}_h, \mathbf{K}_h, \mathbf{V}_h\right) = \text{softmax}\left(\frac{\mathbf{Q}_h\mathbf{K}_h^\top}{\sqrt{D}}\right)\mathbf{V}_h \tag{4}$$

The symbol $\|$ denotes the concatenation of multiple attention heads, and $\mathbf{W}^O$ is the projection matrix that aligns the output dimensions with those of $\mathbf{H}$. Here, we let variables $a$ and $b$ indicate spatial ($s$) indicator and main observation ($o$), i.e., $a = s$ and $b = o$. Then the queries, keys, and values are generated through following transformations,

$$\mathbf{Q}_h^{(s)} = \mathbf{H}[\ldots, \text{slice}^{(s)}]\mathbf{W}_h^{(Q_s)}, \quad \mathbf{K}_h^{(o)} = \mathbf{H}[\ldots, \text{slice}^{(o)}]\mathbf{W}_h^{(K_o)}, \quad \mathbf{V}_h^{(o)} = \mathbf{H}[\ldots, \text{slice}^{(o)}]\mathbf{W}_h^{(V_o)} \tag{5}$$

where $\mathbf{W}$ transforms input data into dimension $D$ of attention space, slice$^{(s)}$ and slice$^{(o)}$ denote the respective slices of $\mathbf{H}$ for spatial and observational features. After computing attention scores, the embeddings are taken into a feed-forward network (FFN) to enhance the learning capabilities, where $\text{FFN}(\mathbf{X}) = \max(0, \mathbf{X}\mathbf{W}_1 + \mathbf{b}_1)\mathbf{W}_2 + \mathbf{b}_2$. The final attention outputs are then normalized by,

$$\begin{aligned}
\tilde{\mathbf{H}}^{(a,b)}[\ldots, \text{slice}^{(b)}] &= \text{LN}(\text{FFN}(\text{LN}(\text{MHCA}^{(a,b)}(\mathbf{H}) + \mathbf{H}[\ldots, \text{slice}^{(b)}])) \\
&\quad + \text{LN}(\text{MHCA}^{(a,b)}(\mathbf{H}) + \mathbf{H}[\ldots, \text{slice}^{(b)}])),
\end{aligned} \tag{6}$$

where LN is layer normalization, the resulting matrices $\tilde{\mathbf{H}}^{(s,o)}$ and $\tilde{\mathbf{H}}^{(o,s)}$ are then concatenated back to $\mathbf{H}$ at their respective dimensions as $\tilde{\mathbf{H}}^{(SCCI)} \in \mathbb{R}^{T \times N \times d_h}$, enriching the original representation with refined features that encapsulate intricate cross-dimensional relationships.

**Temporal-context cross-interaction.** To facilitate the attention computation with respect to temporal dimension, the representation is first transposed as $\tilde{\mathbf{H}}'^{(SCCI)} \in \mathbb{R}^{N \times T \times d_h}$ by denoting $T$ as the sequence length for subsequent attention calculations. Then the step-wise positional encoding (refer to Appendix B.2) is introduced to allow our attention aware of specific temporal evolution.

The subsequent steps closely follow those of the spatial-context cross-attention mechanism, so the temporal-context cross-interaction (TCCI) can be performed as $\tilde{\mathbf{H}}^{(CI)} = \text{TCCI}(\tilde{\mathbf{H}}'^{(SCCI)})$, and final representation $\tilde{\mathbf{H}}^{(CI)} \in \mathbb{R}^{N \times T \times d_h}$ becomes the outcome of cross interactions between spatial and temporal dimensions.

**Self-interactions within Spatial and Temporal Aspects.** We proceed to apply self-interaction across different dimensions of the representations using self-attentions [21]. Specifically, we begin with

---

[2]We introduce a task summarization as the prompt (refer Section 4.4) to generate the distinct prompts.

temporal dimension, as the sequence length of representation dimension has naturally been aligned as $T$. This setup allows direct computation of the required self-attention,

$$\text{MHA}(\mathbf{X}) = \Big\|_{h=1}^{\text{head}} \text{Attention}\left(\mathbf{Q}_h, \mathbf{K}_h, \mathbf{V}_h\right) \mathbf{W}^O \tag{7}$$

In this multi-head attention (MHA) configuration, the attention calculation (referred to in Equation 4, 5) involves queries, keys, and values, which is derived from the entire representation $\tilde{\mathbf{H}}^{(CI)}$, rather than individual slices. Then the output undergoes further processing through a FFN and non-linear transformations and LN to stabilize and enrich the feature representations similar Equation 6. The resulting $\tilde{\mathbf{H}}^{(TSI)}$ in $\mathbb{R}^{N \times T \times d_h}$ signifies the outcome of temporal self-interactions (TSI). Then the tensor is transposed to $\tilde{\mathbf{H}}'^{(TSI)} \in \mathbb{R}^{T \times N \times d_h}$. The final spatial self-interaction (SSI) computation is analogous to the temporal version, which refines spatial interactions and aggregates features across spatial nodes, as $\tilde{\mathbf{H}} = \text{SSI}(\tilde{\mathbf{H}}'^{(TSI)})$. The resulting tensor from this computation is $\tilde{\mathbf{H}} \in \mathbb{R}^{T \times N \times d_h}$, which represents the outcome of comprehensive multi-dimensional interactions.

We then adaptively integrate the interactions by a fusion strategy and Huber loss is adopted to ensure the robustness to outliers in spatiotemporal samples. Details can be found in Appendix B.3. Our MSTI allows extracting diverse interactions, including spatial-temporal domain interactions via designing cross attentions on respective indicators, and self interactions within respective dimensions, enhancing the data relation learning and supporting commonality extraction across task domains.

### 4.4 Rolling Adaptation over Continuous Multi-task Spatio-Temporal Learning

To ensure continuous task learning, we propose a Rolling adaptation scheme, RoAda, to model the distinction and commonality among task domains. Our RoAda is composed of two stages with a warm-up for commonality extraction and a task-specific refinement. Before the task rolling, we construct prompts to distinguish personalization of each task. Thus, the commonality and diversity can be leveraged to boost individual task adaptation.

**Task summarization as prompts.** To capture the task distinction, we devise a task summarization by a Sampling-AutoEncoding scheme from each task. Consider task $\mathcal{T}_k$, main observation becomes $\mathbf{X}^{(\mathcal{T}_k)} \in \mathbb{R}^{T_{all}^{(\mathcal{T}_k)} \times N \times C}$. Such data is sampled by averaging observations over equivalent times of day, yielding a periodic sample representation $\mathbf{X}_{samp}^{(\mathcal{T}_k)} \in \mathbb{R}^{L_t^{(\mathcal{T}_k)} \times N \times C}$, where $L_t^{(\mathcal{T}_k)}$ denotes the number of time slots for task $k$ within a day. Since neural networks tend to fit any data regularity, the sampled features are fed into an autoencoder for extracting the compressed and distinguished data patterns. Given the encoding and decoding processes, i.e., $\phi : \mathcal{X}_{samp} \to \mathcal{S}$ and $\psi : \mathcal{S} \to \mathcal{X}'_{samp}$, $\mathbf{S}$ encapsulates the summarized core characteristics of the task. The decoding phase maps $\mathbf{S}$ back to a reconstructed $\mathbf{X}'_{samp}$, by minimizing the mean squared reconstruction error,

$$\phi : \mathbf{S} = \text{sigmoid}(\mathbf{W}_s \mathbf{X}_{samp} + \mathbf{b}_s), \quad \phi, \psi = \underset{\phi, \psi}{\arg\min} \|\mathbf{X}_{samp} - (\psi \circ \phi)\mathbf{X}_{samp}\|^2 \tag{8}$$

where $\mathbf{W}_s$ is the weight, and $\mathbf{b}_s$ is the bias. Following the encoder, the summary features $\mathbf{S}^{(\mathcal{T}_k)}$ are transformed into the $k$-th task prompt $\mathbf{P}^{(\mathcal{T}_k)} = \text{FC}_p(\mathbf{S}^{(\mathcal{T}_k)}; \theta_p)$ with dimension alignment.

**Weight behavior modeling.** The first warm-up stage is designed via weight behavior modeling, which assimilates the regularity from task to task. This process not only adapts the model to new tasks but also solidifies its ability in generalizing across scenarios by capturing task-wise common relations with modeling of model weight behaviors.

We begin with the task $\mathcal{T}_1$ via independently training the model until its performance stabilizes. By denoting $\mathcal{M}$ as the model learned by MSTI, the training phase can be formally described as,

$$\mathbf{P}^{(\mathcal{T}_1)} \xrightarrow[\text{load}]{\text{prompt}} \mathcal{M}, \quad \text{Train}(\mathcal{M}(\mathcal{D}_{train}^{(\mathcal{T}_1)}; \mathbf{W}_{init})) \xrightarrow[\text{convergence}]{\text{until}} \mathbf{W}_c^{(\mathcal{T}_1)} \tag{9}$$

where $\mathbf{W}_{init}$ are the initialized weights, $\mathcal{D}_{train}^{(\mathcal{T}_1)}$ is the training dataset of task $\mathcal{T}_1$, and $\mathbf{W}_c^{(\mathcal{T}_1)}$ are the weights when model converges. After that, our model transitions learning task from $\mathcal{T}_1$ to $\mathcal{T}_2$ by loading the corresponding task prompt $\mathbf{P}^{(\mathcal{T}_2)}$ and dataset $\mathcal{D}_{train}^{(\mathcal{T}_2)}$. This transition involves a critical step where the evolution behavior of model weights $\mathbf{W}$ are carefully stored,

$$\text{Train}(\mathcal{M}(\mathcal{D}_{train}^{(\mathcal{T}_2)}; \mathbf{W}_c^{(\mathcal{T}_1)})) \xrightarrow[\text{each epoch}]{\text{for}} \mathbf{W}_0^{(\mathcal{T}_2)}, \mathbf{W}_1^{(\mathcal{T}_2)}, \ldots, \mathbf{W}_c^{(\mathcal{T}_2)} \tag{10}$$

To capture common patterns, we reflect the task-level stability and variations by the evolution behavior of weights in $\mathcal{M}$, i.e., $\mathcal{W}^{(\mathcal{T}_2)} = \{\mathbf{W}_c^{(\mathcal{T}_1)}, \mathbf{W}_0^{(\mathcal{T}_2)}, \mathbf{W}_1^{(\mathcal{T}_2)}, \ldots, \mathbf{W}_c^{(\mathcal{T}_2)}\}$. The weight set $\mathcal{W}^{(\mathcal{T}_2)}$ is deliberately incorporated with finalized weight of task $\mathcal{T}_1$ and evolution of $\mathcal{T}_2$, which explicitly captures the weight transition between tasks. We then introduce a collective variance $\sigma$ to capture such stability, and employ a threshold $\delta$ to disentangle the stable and dynamic weights along the learning process, i.e.,

$$\mathbf{W}_{\text{stable}}^{(\mathcal{T}_2)}, \mathbf{W}_{\text{dynamic}}^{(\mathcal{T}_2)} = \{w_{ij} \in \mathcal{W}^{(\mathcal{T}_2)} : \text{Var}(w_{ij}) < \delta\}, \{w_{ij} \in \mathcal{W}^{(\mathcal{T}_2)} : \text{Var}(w_{ij}) \geq \delta\} \quad (11)$$

where $\text{Var}(w_{ij})$ represents the element-wise variance of the across different training iterations from $\mathbf{W}_c^{(\mathcal{T}_1)}$ to $\mathbf{W}_c^{(\mathcal{T}_2)}$, indicating the fluctuation degree of the weight values. A lower variance indicates higher stability, suggesting minimal change in weights across updates. After that, stable weights $\mathbf{W}_{\text{stable}}^{(\mathcal{T}_2)}$ are then frozen, and the model transitions to the next task, $\mathcal{T}_3$, using the stabilized weights as the initialization for further training,

$$\text{Train}(\mathcal{M}(\mathcal{D}_{train}^{(\mathcal{T}_3)}; \mathbf{W}_c^{(\mathcal{T}_2)}.\text{frozen}(\mathbf{W}_{\text{stable}}^{(\mathcal{T}_2)}))) \xrightarrow[\text{each epoch}]{\text{for}} \mathbf{W}_0^{(\mathcal{T}_3)}, \mathbf{W}_1^{(\mathcal{T}_3)}, \ldots, \mathbf{W}_c^{(\mathcal{T}_3)} \quad (12)$$

Similar to Equation 12, this process is repeated until the completion of task $\mathcal{T}_k$, resulting in the collection $\mathcal{W}^{(\mathcal{T}_k)} = \{\mathbf{W}_c^{(\mathcal{T}_{k-1})}, \mathbf{W}_0^{(\mathcal{T}_k)}, \mathbf{W}_1^{(\mathcal{T}_k)}, \ldots, \mathbf{W}_c^{(\mathcal{T}_k)}\}$. The Equation 11 is similarly implemented to derive $\mathbf{W}^{(\mathcal{T}_k)} = \mathbf{W}_{\text{stable}}^{(\mathcal{T}_k)} \parallel \mathbf{W}_{\text{dynamic}}^{(\mathcal{T}_k)}$. Since $\mathcal{T}_1$ is not involved with the commonality extraction, we subsequently load $\mathcal{T}_1$ with $\mathbf{W}^{(\mathcal{T}_k)}.\text{frozen}(\mathbf{W}_{\text{stable}}^{(\mathcal{T}_k)})$ to achieve the complete rolling process. The stabilized weights across continuous tasks $\mathbf{W}'^{(\mathcal{T}_1)} = \mathbf{W}'^{(\mathcal{T}_1)}_{\text{stable}} \parallel \mathbf{W}'^{(\mathcal{T}_1)}_{\text{dynamic}}$, serves as $\mathbf{W}^*$ can ultimately result in a robust multi-task learning parameters encapsulated with well-extracted common patterns via iterative stable weight selection, and it can also be served a collective intelligence by exploiting multiple tasks, thus enhancing the generalization for subsequent learning.

**Task-specific refinement phase.** This phase aims to merge the gap between task-level commonality and specificity, where the process continuously train the model from the prior phase to the next one, as $\mathbf{W}^*_{\text{stable}}$ is frozen to maintain overall model stability and remaining weights are iteratively update with task-specific prompts. This process allows CMuST to insert the individual intelligence into integrated model and adjust itself to better suit the unique pattern of each task, i.e.,

$$\mathcal{M}^{(\mathcal{T}_i)} = \text{FineTuning}(\mathcal{M}, \mathbf{W}^*, P^{(\mathcal{T}_i)}) \quad (13)$$

where $\mathcal{M}^{(\mathcal{T}_i)}$ is tuned to maximize performance on the task at hand, which represents the task-specific submodel after refinement. The FineTuning denotes adjustment process of model parameters.

**Summary.** Our RoAda not only ensures the preservation of commonalities across tasks, but each model is also optimally tuned for its respective task with compressed task-level patterns, providing CMuST with opportunity to achieve peak performance with both collective and individual intelligence. More details of our methodology can be found in Appendix.

## 5 Experiment

### 5.1 Datasets and Experiment Setup

**Data description.** Given the emerging multi-task ST learning, we collect and process three real-world datasets for evaluation: 1) **NYC**[3]: Includes three months of crowd flow and taxi hailing from Manhattan and its surrounding areas in New York City, encompassing four tasks: *Crowd In*, *Crowd Out*, *Taxi Pick*, and *Taxi Drop*. 2) **SIP**: Contains records of *Traffic Flow* and *Traffic Speed* within Suzhou Industrial Park over a period of three months. 3) **Chicago**[4]: Comprises of traffic data collected in the second half of 2023 from Chicago, including three tasks: *Taxi Pick*, *Taxi Drop*, and *Risk*.

**Baselines and metrics.** Our CMuST model is evaluated on a widely-used baselines spatiotemporal prediction, including RNN-based models (**DCRNN** [19], **AGCRN** [4]), STGNNs (**GWNET** [40], **STGCN** [44]) *for single task*, and **MTGNN** [39], **STEP** [33] **PromptST** [48] *for multiple tasks*,

---

[3]https://www.nyc.gov/site/tlc/about/tlc-trip-record-data.page
[4]https://data.cityofchicago.org/browse

Table 1: Performance comparison on three datasets. Best results are **bold** and the second best are underlined.

| Datasets | | NYC | | | | SIP | | Chicago | | |
|---|---|---|---|---|---|---|---|---|---|---|
| Methods | Metrics | Crowd In | Crowd Out | Taxi Pick | Taxi Drop | Traffic Flow | Traffic Speed | Taxi Pick | Taxi Drop | Risk |
| DCRNN | MAE | 17.5289 | 19.5667 | 10.8188 | 9.6142 | 12.5326 | 0.7044 | 3.0624 | 2.5793 | 1.1174 |
| | MAPE | 0.5939 | 0.5695 | 0.4330 | 0.4818 | 0.2455 | 0.2686 | 0.4237 | 0.4816 | 0.2504 |
| AGCRN | MAE | 11.5135 | 13.1569 | 7.0675 | 6.0066 | 15.8319 | 0.6924 | 2.3542 | 2.0884 | 1.1183 |
| | MAPE | 0.5094 | 0.4773 | 0.3753 | 0.3665 | 0.2926 | 0.2744 | 0.4092 | 0.4046 | 0.2505 |
| GWNET | MAE | 11.4420 | 13.2992 | 7.0701 | 6.1171 | 13.0529 | 0.6900 | 2.3671 | 2.0434 | 1.1197 |
| | MAPE | 0.4778 | 0.6171 | 0.3713 | 0.3514 | 0.2483 | 0.2655 | 0.3912 | 0.4044 | 0.2514 |
| STGCN | MAE | 11.3766 | 13.3522 | 7.1259 | 5.9268 | 15.3501 | 0.7111 | 2.3781 | 2.1427 | 1.1184 |
| | MAPE | 0.5018 | 0.4318 | 0.3234 | **0.3339** | 0.3041 | 0.2660 | 0.4074 | 0.4331 | 0.2507 |
| GMAN | MAE | 11.3414 | 13.1923 | 7.0662 | 6.0912 | 13.0368 | 0.6952 | 2.3663 | 2.0316 | 1.1182 |
| | MAPE | 0.4782 | 0.6065 | 0.3652 | 0.3468 | 0.2464 | 0.2678 | 0.3953 | 0.4036 | 0.2516 |
| ASTGCN | MAE | 14.2847 | 17.1582 | 9.1430 | 7.7063 | 16.4896 | 0.6980 | 2.5091 | 2.1520 | 1.1175 |
| | MAPE | 0.6396 | 0.5922 | 0.4607 | 0.4524 | 0.3104 | 0.2682 | 0.4593 | 0.4413 | **0.2502** |
| STTN | MAE | 12.1994 | 14.1966 | 7.6716 | 6.3816 | 15.1751 | 0.6939 | **2.2996** | 2.0355 | 1.1214 |
| | MAPE | 0.4757 | 0.4744 | 0.3600 | 0.3763 | 0.2881 | 0.2625 | 0.3893 | 0.4133 | 0.2518 |
| MTGNN | MAE | 11.4350 | 13.3072 | 7.0736 | 6.1162 | 13.0486 | 0.6989 | 2.3692 | 2.0361 | 1.1201 |
| | MAPE | 0.4785 | 0.6185 | 0.3782 | 0.3502 | 0.2475 | 0.2687 | 0.3979 | 0.4073 | 0.2578 |
| STEP | MAE | 11.2328 | 13.1043 | 6.9619 | 5.9101 | 12.0032 | 0.6970 | 2.3592 | 2.0168 | 1.1190 |
| | MAPE | 0.4537 | 0.4361 | 0.3248 | 0.3379 | 0.2391 | 0.2638 | 0.3914 | 0.4019 | 0.2507 |
| PromptST | MAE | **11.0036** | 13.0237 | 6.8711 | 5.8797 | 11.8620 | 0.6921 | 2.3576 | 2.0065 | 1.1186 |
| | MAPE | 0.4465 | 0.4358 | 0.3265 | 0.3382 | 0.2375 | 0.2632 | 0.3913 | 0.4012 | 0.2511 |
| CMuST | MAE | 11.1533 | **12.9088** | **6.7581** | **5.8546** | **11.5811** | **0.6843** | 2.3264 | **2.0034** | **1.1172** |
| | MAPE | **0.4384** | **0.4265** | **0.3118** | 0.3375 | **0.2279** | **0.2585** | **0.3872** | 0.4009 | 0.2503 |

as well as _attention-based models_ of (**GMAN** [50], **ASTGCN** [9], **STTN** [42]). The performance metrics are mean absolute error (MAE), and mean absolute percentage error (MAPE), where lower values indicate higher predictive performance.

**Implementation details.** We partitioned datasets into training, validation, and testing sets with 7:1:2 ratio. CMuST forecasts observations of next 12 time steps based on previous 12 steps, as $T = T' = 12$, . All data were normalized to zero mean and unit variance, and predictions were denormalized to normal values for evaluation. For the MSTI, embedding dimensions were $d_{obs} = 24, d_s = 12, d_t = 60$, and the prompt dimension was 72. Dimensions for self-attention and cross-attention respectively were 168 and 24, with each attention having 4 heads and FFN's hidden dimension was 256. The Adam optimizer is adopted with an initialized learning rate of $1 \times 10^{-3}$, and weight decay of $3 \times 10^{-4}$, where the early-stop was applied. For RoAda, the threshold $\delta = 10^{-6}$. Our model was implemented with PyTorch on a Linux system equipped with Tesla V100 16GB.

## 5.2 Performance Comparison

1) **Performance comparison among baselines.** In Table 1, we compared the predictive performance of our CMuST with other methods across various city datasets. Obviously, our CMuST significantly outperforms other baselines on all datasets across most of metrics and the multitask-based methods improve performance by an average of 8.51% over the singletask-based methods. This result underscores the effectiveness of the cross-attention mechanism for decoupling multidimensional dependencies, which not only enhances spatiotemporal representational capacity, but also enables easy extraction of common correlations among tasks and thus empowering each individual task to benefit from well-extracted common patterns.

Table 2: Performance against data sparsity.

| model | NYC for Crowd In | | | | | | | |
|---|---|---|---|---|---|---|---|---|
| | 25% nodes | | 50% nodes | | 2 times interval | | 4 times interval | |
| | MAE | MAPE | MAE | MAPE | MAE | MAPE | MAE | MAPE |
| GWNET | 13.7648 | 0.4825 | 12.4637 | 0.4731 | 20.2547 | 0.4465 | 20.6487 | 0.4958 |
| STEP | 13.1827 | 0.4772 | 12.2393 | 0.4612 | 20.1936 | 0.4436 | 20.1465 | 0.4915 |
| PromptST | 12.8362 | 0.4719 | 12.0361 | 0.4607 | 19.8465 | 0.4384 | 19.5238 | 0.4872 |
| CMuST | **12.1611** | **0.4506** | **11.2864** | **0.4470** | **18.2925** | **0.4279** | **18.4084** | **0.4797** |

2) **Robustness in data-scarce scenarios within multi-task framework.** We constructed scenarios of data scarcity for specific tasks, i.e., we reduce part of the spatial nodes in prediction of _Crowd In_, and also reduce number of samples in temporal dimension by expanding time intervals on NYC, to study the robustness of CMuST under the challenging scenario of limited (reduced) data. Results shown in Table 2 indicate that assimilating common information from other tasks can help better prediction in a single task even though it is with limited data on either spatial and temporal dimension. This demonstrates that multi-task prediction relax the requirement of individual tasks on both data volumes and distributions, where the shared commonality effectively captures and delivers the consistency and diversity among tasks.

## 5.3 Ablation Study

To assess the effectiveness of each module in CMuST and its capability for multi-task learning, we designed a set of variants as 1) **w/o context-data interaction:** remove the spatial-context and temporal-context cross-interactions in the MSTI module, 2) **w/o consistency maintainer:** omit the separation and freezing of stable weights during the RoAda phase, instead using all weights for rolling training, 3) **w/o task-specific preserver:** eliminate the task-specific prompts, thus removing the task-specific diversity preservation. Figure 3 supports our hypothesis that CMuST is a cohesive and integral system, where the results of "w/o context-data interaction" deteriorate,

potentially indicating multidimensional interactions from context environments to spatial relations and temporal evolution exactly make sense for prediction. The results on "w/o consistency maintainer" highlights the validity of capturing task consistency and commonality to facilitate task learning. Lastly, performances of "w/o task-specific preserver" show that removing unique patterns distinctive from other tasks makes inferior results.

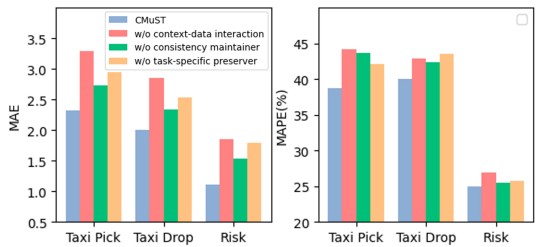

Figure 3: Ablation studies of CMuST on Chicago.

## 5.4 Case Study

1) **Visualizing attention across training phases.** In Figure 4(a), we visualize the changes of attention weights of CMuST during the stage of RoAda. It is observed that as tasks are learning continuously, the relationships and interactions across various dimensions is becoming distinctive and going stable, demonstrating the consolidation process of dimension-level relations. By modeling weight behavior, such consolidated relations and interactions between context and observations can further enable the extraction of consistency in spatiotemporal interactions across tasks. 2) **Performance variation along with task increasing.** Figure 4(b) shows the performance of individual tasks on NYC as the number of tasks increases. The performance of each task is improving with the addition of more tasks, which indicates that tasks are no longer isolated but gain the collective intelligence via assimilating common representations and interactive information.

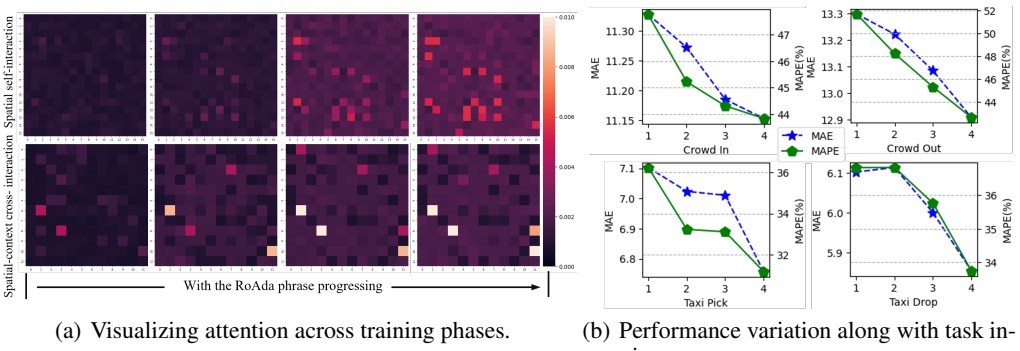

(a) Visualizing attention across training phases.

(b) Performance variation along with task increasing.

Figure 4: Case studies of our proposed CMuST on NYC.

## 5.5 Parameter sensitivity analysis

We varied the dimension of the task prompt $d_p$ as $\{18, 36, 72, 144\}$, the number of attention heads in MSTI as $\{1, 2, 4, 8, 16\}$, and the threshold $\delta$ for RoAda among $\{10^{-4}, 10^{-5}, 10^{-6}, 10^{-7}\}$. Results shown in Figure 6 indicate that the optimal settings were $d_p = 72$, $head = 4$, and $\delta = 10^{-6}$. Details of parameter analysis and fine-tuning can be found in our Appendix C.6.

## 6 Conclusion

In this work, we enable spatiotemporal learning to get rid of isolation with proposed CMuST, which consists of two major components. In CMuST, the MSTI is devised to dissect complex multi-dimension data correlations, to reveal disentangled patterns. To extract the task-wise consistency

and task-specific diversity, we propose a rolling learning scheme RoAda, where it simultaneously models weight behavior to enable collective intelligence, and constructs task-specific prompts by compressing domain data with AutoEncoding to empower task-specific refinement for enhancement. We believe our CMuST can not only help better understand the collective regularity and intelligence in urban systems, but significantly reduce repeated training and improve data exploitation, which is progressively approaching green computing in future cities. For future work, we will further investigate the collective intelligence in open urban systems, which can potentially generalize to wider domains such as energy and environment for human-centered computing.

## 7 Acknowledgement

This paper is partially supported by the National Natural Science Foundation of China (No.12227901, No.62072427), Natural Science Foundation of Jiangsu Province (BK.20240460), the Project of Stable Support for Youth Team in Basic Research Field, CAS (No.YSBR-005), and the grant from State Key Laboratory of Resources and Environmental Information System.

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

# A Explanation of Relevant Concepts

## A.1 The Concept, Definition and Scope of Multi-task Learning

Actually, in our study, various domains correspond to different urban elements collected with different manners in a given city. For instance, in an integrated urban system, it includes taxi demands, crowd flow, traffic speed and accidents. We collect and organize various domain data (urban elements) in a city into one integrated dataset. The goal of our work is to explore the integrated intelligence from various domains and enhance learning of each individual urban element. To this end, the concept of multi-task here is to forecast various elements from different domains in an integrated model. Therefore, our work does not target at unifying regression or classification problems, but proposes an integrated model to iteratively establish the common intelligence among different elements and improve generalization for each element learning in succession, thus getting rid of task isolation. Noted that our experiments are performed with regression tasks, but it can easily generalize to classification task with shared representations.

## A.2 Continuous & Continual learning

In this work, 'continuous' is equivalent to 'continual'. The uniqueness of our work refers to a novel continuous task learning in ST community, which collects the integrated intelligence and benefits each individual learning task.

# B Methodology Details

## B.1 Illustration of Data Representation and Integration

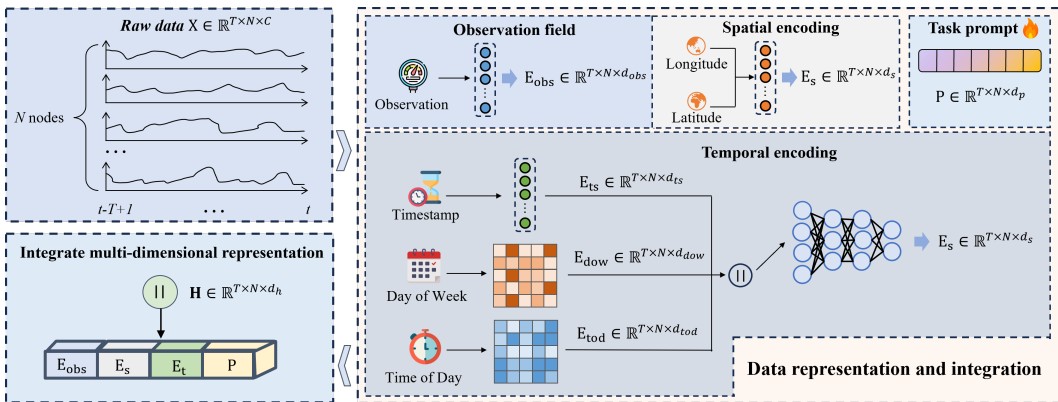

Figure 5: This figure illustrates the process of Data Representation and Integration within the CMuST framework. $T$ and $N$ denote the length of the time series and the number of geographic nodes, respectively. The diagram on the right details the encoding of information across different dimensions, with the bottom right corner aggregating the results of all encodings into a unified multidimensional representation.

## B.2 Positional Encoding for Temporal-context Cross-interaction

Given the $D$ dimensions in total, we define the positional encoding as,

$$
\begin{cases}
\text{pos}_{(t,2d)} = \sin\left(t/10000^{2d/D}\right) \\
\text{pos}_{(t,2d+1)} = \cos\left(t/10000^{2d/D}\right)
\end{cases}
\qquad \text{for } d = 0, \cdots, D/2 - 1
\tag{14}
$$

where $t$ and $d$ are the indexes of time slots and feature dimensions, which alternately uses sine and cosine to provide a unique representation for each time slot.

### B.3 Fusion & Regression

Predictions for future steps are influenced by representations affected by multi-dimensional interactions, where the impact of each dimension may vary. Therefore, we integrate information across various aspects effectively by a parameter matrix-based fusion strategy,

$$\mathbf{Z} = \mathbf{W}_o * \tilde{\mathbf{H}}[\ldots, \text{slice}^{(o)}] + \mathbf{W}_s * \tilde{\mathbf{H}}[\ldots, \text{slice}^{(s)}] + \mathbf{W}_t * \tilde{\mathbf{H}}[\ldots, \text{slice}^{(t)}] \qquad (15)$$

where $*$ denotes the convolution operation, employing 1 x 1 convolution kernels. $\mathbf{W}_o$, $\mathbf{W}_s$, and $\mathbf{W}_t$ are the parameters within these kernels, tailored to adjust the influence of observational data, spatial locations, and temporal information on the prediction targets, respectively. We further integrate the task-specific prompt to culminate in the final prediction $\hat{\mathbf{Y}} = \text{FC}_y(\mathbf{Z}\mathbf{W}_z + \tilde{\mathbf{H}}[\ldots, \text{slice}_p]\mathbf{W}_p; \theta_y)$, the Huber Loss [26] is utilized as the optimization function, which is less sensitive to outliers compared to squared error loss, defined as:

$$\mathcal{L}_y = \begin{cases} \frac{1}{2}(\hat{\mathbf{Y}} - \mathbf{Y})^2, & \text{if } |\hat{\mathbf{Y}} - \mathbf{Y}| < \delta \\ \delta(|\hat{\mathbf{Y}} - \mathbf{Y}| - \frac{1}{2}\delta), & \text{otherwise} \end{cases} \qquad (16)$$

where $\mathbf{Y}$ is the ground truth, $\delta$ controls the sensitivity to outliers, which enhances the accuracy by integrating diverse data dimensions and improves robustness against data variability and anomalies.

### B.4 Details of Rolling Adaptation

The detailed algorithmic procedure for the RoAda phase can be found in Algorithm 1.

## C Additional Experiment Details

### C.1 Dataset Details

We provide detailed information about the dataset, including the number of records in the original data, the number of regions into which the data was divided, and the time intervals. These details are presented in Table 3. The specific data preprocessing is as follows:

**NYC**: We collect yellow taxi trip data from January to March 2016 from the NYC Open Data website. Each trip record includes information such as pickup and dropout times, locations, and the number of passengers. We filter out records with abnormal longitude and latitude values or missing data. Then we select data within Manhattan and surrounding areas, divided into 30x15 grids, and counted trips per grid, selecting those with total trips greater than or equal to 1000, resulting in 206 grids. Each grid's data is aggregated into 30-minute intervals, yielding taxi pickup counts, taxi dropout counts, and crowd in/out flows. We also include time of day (tod) and day of week (dow) as context, resulting in four tasks with input features [value, tod, dow].

**SIP**: We collect traffic data from Suzhou Industrial Park from January to March 2017, comprising tens of thousands of records. The area is divided into nodes, and data is aggregated into 5-minute intervals. After filtering out grids with sparse data, we obtain 108 nodes, each containing traffic speed and traffic flow. We include time of day and day of week as input context, resulting in two tasks: traffic flow and traffic speed, with input [value, tod, dow].

**Chicago**: We collect taxi trip and accident data from the Chicago Open Data platform for June to December 2023. The taxi data includes trip start, end times and locations. We divide the area into 30x20 grids and select grids with total trips greater than 100, resulting in 220 grids. Similar to the NYC dataset, data is aggregated into 30-minute intervals, yielding taxi pickup and dropout counts, resulting in two tasks with input features [value, tod, dow]. The accident data includes incident locations, times, casualty numbers, and injury severity of each casualty. We then obtain the risk score by weighting it according to each casualty and injury, mapped it to the 220 grids, and aggregated the risk score over time intervals, resulting in a risk task with input features [risk score, tod, dow].

### C.2 Implementation Details and Fairness-aware Experimental Evaluation

To verify whether the multi-task learning in urban systems can compete single-task learning scheme and further show the superiority of our continuous multi-task learning, our experiments are designed

---

**Algorithm 1** Rolling Adaptation Process

---

**Require:** Model $\mathcal{M}$, tasks $\{\mathcal{T}_1, \mathcal{T}_2, \ldots, \mathcal{T}_k\}$, prompts $\{\mathbf{P}^{(\mathcal{T}_1)}, \mathbf{P}^{(\mathcal{T}_2)}, \ldots, \mathbf{P}^{(\mathcal{T}_k)}\}$, train data
$\qquad \{\mathcal{D}_{\text{train}}^{(\mathcal{T}_1)}, \mathcal{D}_{\text{train}}^{(\mathcal{T}_2)}, \ldots, \mathcal{D}_{\text{train}}^{(\mathcal{T}_k)}\}$, learning rate $\gamma$, threshold $\delta$

1: **Warm-up Phase:**
2: ▷ *For all parameters in $\mathcal{M}$, the parameters corresponding to their respective names* ◁
3: **Initialize:** $\mathcal{W}_{histories} \leftarrow \{\}$  ▷ *Dictionary with keys as names and values as parameters list*
4: **for** each task $\mathcal{T}_i$ in $\{\mathcal{T}_1, \mathcal{T}_2, \ldots, \mathcal{T}_k\}$ **do**
5:   **if** $i == 1$ **then**
6:     ▷ *Train task 1 for initializing model* ◁
7:     Load prompt for task $\mathcal{T}_1$: $\mathcal{M} \leftarrow \mathbf{P}^{(\mathcal{T}_1)}$
8:     Set learning rate as $\gamma$
9:     Train $\mathcal{M}$ on $\mathcal{D}_{\text{train}}^{(\mathcal{T}_1)}$ until convergence to obtain $\mathbf{W}_c^{(\mathcal{T}_1)}$
10:     Store weights: $\mathcal{W}_{histories} \xleftarrow[\text{name:parameters}]{\text{append}} \mathbf{W}_c^{(\mathcal{T}_1)}$
11:   **else**
12:     ▷ *Rolling training for task 2 to k* ◁
13:     Load prompt for task $\mathcal{T}_i$: $\mathcal{M} \leftarrow \mathbf{P}^{(\mathcal{T}_i)}$
14:     Set learning rate as $\gamma \times 0.01$    ▷ *Prevent catastrophic forgetting during rolling*
15:     Train $\mathcal{M}$ on $\mathcal{D}_{\text{train}}^{(\mathcal{T}_i)}$ with initial weights $\mathbf{W}_c^{(\mathcal{T}_{i-1})}$
16:     **for** each epoch $j$ **do**
17:       Record weights: $\mathcal{W}_{histories} \xleftarrow[\text{name:parameters}]{\text{append}} \mathbf{W}_j^{(\mathcal{T}_i)}$
18:     After training, obtain the weights of $\mathcal{M}$ as $\mathbf{W}_c^{(\mathcal{T}_i)}$
19:     ▷ *Calculate variances and freeze stable weights* ◁
20:     **for** each parameters' name in $\mathcal{M}$ **do**
21:       $v \leftarrow$ CALCULATE_VARIANCE($\mathcal{W}_{histories}[name]$)
22:       **if** $v < \delta$ **then**
23:         Freeze the parameter corresponding to the name
24:     Reset $\mathcal{W}_{histories}$ with $\mathbf{W}_c^{(\mathcal{T}_i)}$
25: ▷ *Train task 1 again since it's not involved with the commonality extraction* ◁
26: $\mathcal{M} \leftarrow \mathbf{P}^{(\mathcal{T}_1)}$
27: The remaining steps are similar to lines 14-22
28: Finally, the weights of $\mathcal{M}$ are saved as $\mathbf{W}^*$
29: **Task-Specific Refinement Phase:**
30: **for** each task $\mathcal{T}_i$ in $\{\mathcal{T}_1, \mathcal{T}_2, \ldots, \mathcal{T}_k\}$ **do**
31:   Load weights and prompt $\mathcal{M} \leftarrow \mathbf{W}^*, \mathcal{M} \leftarrow \mathbf{P}^{(\mathcal{T}_i)}$
32:   Set learning rate as $\gamma$
33:   Fine-tune $\mathcal{M}$ on $\mathcal{D}_{\text{train}}^{(\mathcal{T}_i)}$ and save model

---

into both single-task learning and multi-task learning. We will further clarify how to ensure the fair evaluation and comparisons across different models.

For single-task learning, we take different datasets as individual ones to respectively train and test the model [23], where compared models and our CMuST are respectively and repeatedly trained for 5 times with different random seeds. The average results are reported in our Table. 1.

For multi-task learning, since existing models have not designed the task-level continuous learning scheme, we align the features of different types of data in the same urban system into a same urban graph, where data features are concatenated for training the model. The learning objectives are followed by corresponding statement of each paper. For our CMuST, we implement the continuous and iterative model update described in Sec. 4.4. The fairness is well-incorporated by ensuring the all the data input to each model is equivalent.

The experimental results find that almost all multi-task learning can compete against single-task learning, and our continuous multi-task learning is also superior to other multi-task learning without explicitly capturing the task commonality and diversity.

Table 3: The detailed information of the three datasets.

| City | Task | #Records | Time Span | #Regions | #Time Steps | Time Interval |
|---|---|---|---|---|---|---|
| **NYC** | Taxi Drop Taxi Pick Crowd In Crowd Out | 30,245k | 01/01/2016- 03/31/2016 | 206 | 4368 | 30mins |
| **SIP** | Traffic Flow Traffic Speed | 1,237k 307k | 01/01/2017- 03/31/2017 | 108 | 25920 | 5mins |
| **Chicago** | Taxi Drop Taxi Pick Risk | 3,291k 61k | 06/01/2023- 12/31/2023 | 220 | 10272 | 30mins |

## C.3 Avoiding Catastrophic Forgetting

To avoid catastrophic forgetting, we implement several strategies during the RoAda phase. Firstly, we set the learning rate for each task to 1e-5. This helps to retain more knowledge from previous tasks and prevents the model from over-adjusting to new tasks. By maintaining a low learning rate, the model can incrementally learn new information while preserving the stability of previously learned tasks. Additionally, we use task-specific weights for each task, such as task prompts. This method allows us to absorb the common features across all tasks while independently preserving and updating task-specific parameters. This approach ensures that when learning new tasks, the model does not forget the knowledge gained from previous tasks, thereby preventing catastrophic forgetting and avoiding overfitting to new tasks.

## C.4 Comparison Experiments with Unified Models

We compare with unified spatiotemporal/time series learning, such as UniST [45] and UniTime [22], to better show the generalization ability of our model for multi-task learning in the same urban system. The results of these in Table 5.

Table 4: Explanation of table symbols of Table 5 and Table 6. The datasets are numbered by ①②③ respectively, and the specific tasks in the dataset are numbered by Roman numerals III and so on.

| Dataset | Task | Notation(Dataset/task) |
|---|---|---|
| NYC | Crowd In | ①/I |
| NYC | Crowd Out | ①/II |
| NYC | Taxi Pick | ①/III |
| NYC | Taxi Drop | ①/IV |
| SIP | Traffic Flow | ②/I |
| SIP | Traffic Speed | ②/II |
| Chicago | Taxi Pick | ③/I |
| Chicago | Taxi Drop | ③/II |
| Chicago | Risk | ③/III |

Table 5: The results of unified model.

| Model/Dataset | ①/I | ①/II | ①/III | ①/IV | ②/I | ②/II | ③/I | ③/II | ③/III |
|---|---|---|---|---|---|---|---|---|---|
| UniST/MAE | 11.3865 | 13.0762 | 6.8942 | 5.8804 | 11.7461 | 0.6985 | 2.3551 | 2.0134 | 1.1186 |
| UniST/MAPE | 0.4610 | 0.4478 | 0.3261 | 0.3490 | 0.2465 | 0.2661 | 0.3916 | 0.4162 | 0.2508 |
| UniTime/MAE | 12.2874 | 14.9120 | 7.4723 | 6.4641 | 13.9172 | 0.6993 | 2.4564 | 2.0341 | 1.1292 |
| UniTime/MAPE | 0.4721 | 0.4760 | 0.3671 | 0.3719 | 0.2965 | 0.2713 | 0.3987 | 0.4254 | 0.2511 |

## C.5   Experiments for Cold-start

We have designed the experiment of cold start. Specifically, for NYC dataset, we selected three of the four tasks of *Crowd In*, *Crowd Out*, *Taxi Pick* and *Taxi Drop* in turn for training, and calculated the adaptation time and results for the remaining one task on this basis, comparing with training a single task alone. A similar design is applied for SIP and Chicago datasets. The results are shown in Table 6, which show that, both in terms of effect and time, it performs better than single task, indicating that our model adapts to the newly arrived task more quickly and well, which is conducive to solving the problem of cold start of urban prediction.

Table 6: Experiments for cold start.

| Type/Metric/Dataset | ①/I | ①/II | ①/III | ①/IV | ②/I | ②/II | ③/I | ③/II | ③/III |
|---|---|---|---|---|---|---|---|---|---|
| Single Task/MAE | 11.2457 | 13.1284 | 6.9357 | 6.0122 | 11.8684 | 0.6912 | 2.3317 | 2.0223 | 1.1175 |
| Single Task/MAPE | 0.4623 | 0.4782 | 0.3453 | 0.3531 | 0.2758 | 0.2613 | 0.3983 | 0.4023 | 0.2506 |
| Adaptation Time (s) | 2,132 | 1,344 | 1540 | 1,545 | 4,272 | 4,525 | 3,301 | 4,504 | 4,313 |
| Cold Start/MAE | **11.1681** | **13.0027** | **6.8032** | **5.9834** | **11.7832** | **0.6901** | **2.3278** | **2.0089** | **1.1173** |
| Cold Start/MAPE | **0.4407** | **0.4323** | **0.3225** | **0.3463** | **0.2469** | **0.2598** | **0.3906** | **0.4013** | **0.2504** |
| Adaptation Time (s) | **1,571** | **1,150** | **1,322** | **1,441** | **3,019** | **4,053** | **2,886** | **2,736** | **4,198** |

## C.6   Parameter Sensitivity Analysis

To study the impact of hyperparameters on model performance, we varied the dimension of the task prompt $d_p$ as $\{18, 36, 72, 144\}$, the number of attention heads in MSTI as $\{1, 2, 4, 8, 16\}$, and the threshold $\delta$ for RoAda among $\{10^{-4}, 10^{-5}, 10^{-6}, 10^{-7}\}$. The experimental results, displayed in Figure 6, are based on analysis of three datasets. 1) Initially, as the dimension of the prompt increases, both MAE and MAPE decrease, indicating that within a lower range, increasing $d_p$ benefits the model by capturing more task-specific information. However, performance significantly deteriorates after $d_p$ exceeds 72, suggesting that additional dimensions may carry redundant and non-informative elements that introduce noise and prevent the model from effectively learning useful information. 2) On the other hand, as the number of heads increases, model performance initially improves, allowing the attention mechanism to capture more diverse and dimensional information through different heads. Yet, this positive effect reverses after exceeding four heads, indicating that too many heads can lead to information overlap, reducing efficiency and adding unnecessary complexity to the model, thereby degrading performance. 3) During the rolling adaptation phase, both excessively high and low values for the $\delta$ threshold lead to performance degradation. When $\delta$ is set too low, it results in fewer stable weights being retained, which may fail to preserve essential information critical for learning stability. Conversely, setting $\delta$ too high results in more weights being frozen, which restricts the model's flexibility and adaptability, locking in potentially dynamic information and preventing the model from adapting to novel and evolving data patterns effectively.

# D   Others

## D.1   Limitation

Our current work is confined to a single urban system and our experiments are solely within the transportation sector. The exploration into other domains within the same city, such as electricity, transportation, and pollution multi-task learning is still limited. Addressing these limitations will form the basis of our future research.

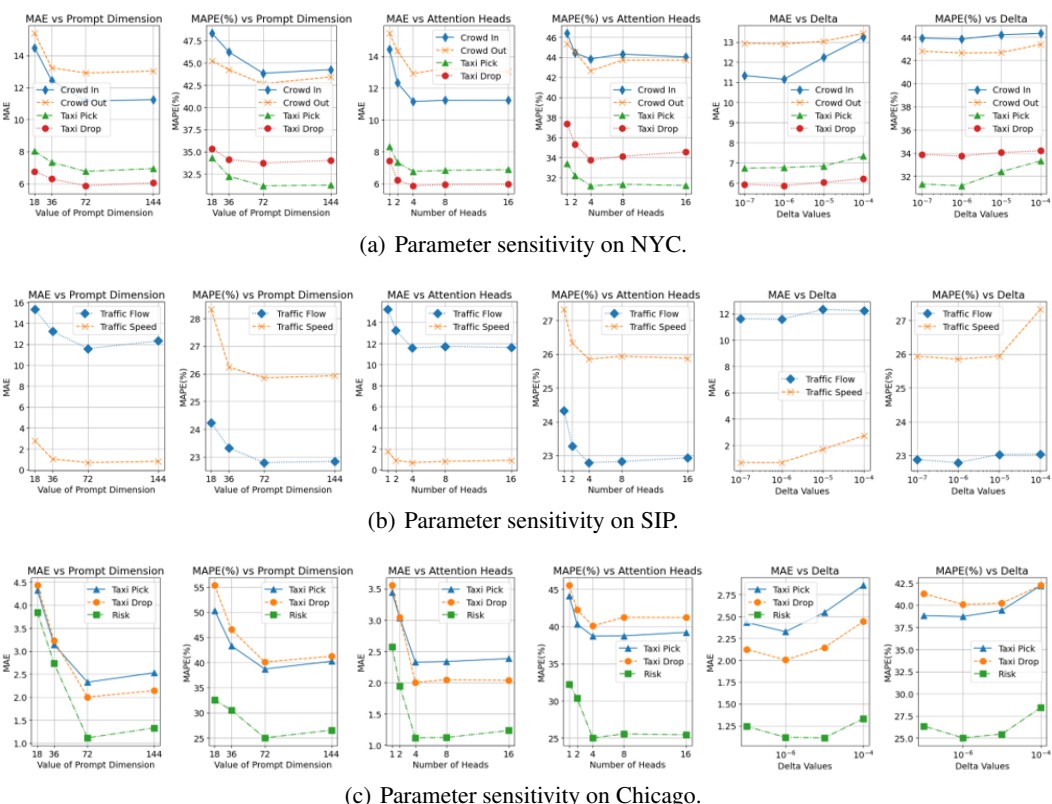

(a) Parameter sensitivity on NYC.

(b) Parameter sensitivity on SIP.

(c) Parameter sensitivity on Chicago.

Figure 6: Parameter sensitivity analysis of CMuST.

