# OpenReview forum: "Get Rid of Isolation: A Continuous Multi-task Spatio-Temporal Learning Framework"
_NeurIPS.cc/2024/Conference — NeurIPS 2024 oral_

### Official Review · Reviewer_xkVJ · 2024-06-23

**Soundness:** 3
**Presentation:** 2
**Contribution:** 1
**Rating:** 5
**Confidence:** 3

**Summary:**

This work proposed a novel spatiotemporal learning framework CMuST. In CMuST, MSTI is devised to dissect complex multi-dimension data correlations, to reveal disentangled patterns. And RoAda is proposed to extract the task-wise consistency and task-specific diversity. In addition, this paper introduce a benchmark of three cities for multi-task spatiotemporal learning, and empirically demonstrate the superiority of CMuST via extensive evaluations on these datasets.

**Strengths:**

1. The paper achieves the best or second-best results in most experiments, validating the feasibility of the proposed method.

2. The paper proposes a continuous learning mechanism that enables the model to continuously learn from tasks. As claimed by the paper, it is "the first continuous multi-task spatiotemporal learning framework, CMuST, to jointly model learning tasks in the same spatiotemporal domain."

**Weaknesses:**

1. The paper considers one of its major contributions to be the proposal of a benchmark. However, after reviewing the code, it seems that the authors did not provide details on how they processed the data.

2. The proposed MTSI module largely uses the Attention module from Transformers, but the authors did not provide any references to Transformers. Additionally, using the attention mechanism to capture relationships is a relatively straightforward design and lacks significant innovation.

3. One of the main problems this paper addresses is the cold start problem for new tasks. However, the paper still involves task-specific refinement, i.e., training is still required. Perhaps the superiority of the proposed module can be validated by comparing the adaptation time to new tasks.

4. In the experiments conducted by the authors, it can be observed that when the number of tasks is one, the model's performance is not superior to many models. This might indicate that the proposed MTSI module is not sufficiently effective.

5. The authors use a simple layer to obtain task summarization $S$. Compared to common designs in many MAEs, using just a linear layer with an activation function seems somewhat simplistic for MAE.

**Questions:**

Please explain each weaknesses.

**Limitations:**

This paper adequately addressed the limitations.

---

> ### Author Rebuttal · Authors · 2024-08-06
>
> Dear Reviewer xkVJ,
>
> Thank you for your meticulous review and insightful comments, which are invaluable in refining our approach and enhancing the quality of our manuscript.
>
> **W1. Data preprocessing.** Thank you for your reminder. We have included the detailed data processing code in our anonymous repository, please check it available, and we explain the preprocessing steps in the **Common Issue 4** of our global response. All datasets and the processed results will be made publicly available on a cloud storage platform after the paper is accepted.
>
> **W2.** **Contribution of MSTI and Transformers references.** In MSTI, our innovation not only lies in designing a new attention mechanism but in how we use it to capture and refine interactions across different dimensions, such as 'spatial aspect - main observation' and 'temporal aspect - main observation' correlations. By coupling with the RoAda process, we enhance capturing  these correlations across multiple tasks, leading to more effective and enriched encoding of spatiotemporal representation over main observations. This multidimensional interaction is specifically designed to exploit the inherent complexities and dependencies in multiple urban elements that standard spatiotemporal models may not fully capture. Additionally, we explain our specific design and the differences from other traditional ST with attention in the **Common issue 2**. Regarding the specific references to the foundational work on Transformers and the attention mechanism, we now have rectified this by including pertinent references to the original works on Transformers by Vaswani et al. [1], as well as other seminal papers that have shaped the use of attention in spatiotemporal learning [2-4], which provides a clearer context for our contributions.
>
> [1] Attention Is All You Need, NeurIPS'17
>
> [2] Learning Dynamics and Heterogeneity of Spatial-Temporal Graph Data for Traffic Forecasting, TKDE'22
>
> [3] STAEformer, CIKM' 23
>
> [4] PDFormer, AAAI'24
>
> **W3. Cold-start problem.** Actually, generalization capacities have been empirically validated in Sec 5.2 and Sec. 5.4, where experiments on Tab. 2 can be viewed as imitating the cold-start issue on spatial dimension. Thanks for your good suggestion, we further add the cold-start experiments concerning tasks, i.e., training on two tasks and testing on another task. The details and results can be found in the **Common issue 3**.
>
> **W4. Performance of single task.** Thanks for your careful reading, we also confirmed this result and found not seriously inferior to other single-task baselines (The comparison  with the baselines have been made over single-task in Tab. 1 of Sec 5.2). Moreover, since in the experiment of Fig. 4(b), the hyperparameters and experimental settings we adopted are based on multi-tasks, investigation over the influence of the task number are also following such multi-task settings to guarantee the fairness of comparisons. This ensures that the only variable is the number of tasks.  Considering learning over individual task, we also believe our backbone, MSTI, can outperform other baselines. To confirm this intuition, we have conducted additional experiments following the individual task setting, (one model for one task), and the testing MAE and MAPE are as follows,
>
> | **Metrics/Dataset** | **①/Ⅰ** | **①/Ⅱ** | **①/Ⅲ** | **①/Ⅳ** | **②/Ⅰ** | **②/Ⅱ** | **③/Ⅰ** | **③/Ⅱ** | **③/Ⅲ** |
> | ------------------- | ------- | ------- | ------- | ------- | ------- | ------- | ------- | ------- | ------- |
> | MAE                 | 11.2457 | 13.1284 | 6.9357  | 6.0122  | 11.8684 | 0.6912  | 2.3317  | 2.0223  | 1.1175  |
> | MAPE                | 0.4623  | 0.4782  | 0.3453  | 0.3531  | 0.2758  | 0.2613  | 0.3983  | 0.4023  | 0.2506  |
>
> (The symbols in the table are explained in the attached **PDF** of global response.)
>
> Even in single-task setting,  comparison with single-task baseline results in Tab. 1, it is observed that  performance of MSTI  is still better than most other single-task models, showing the effectiveness of MSTI. Additionally, the goal of our task is to collective the intelligence across different tasks, and enhance the individual learning, especially improving resilience on extreme scenarios. For detailed evaluation, it include experiments of Sec 5.2, experiments in Sec 5.4, and additional cold-start challenge investigated in our global rebuttal (Results in **Common issue 3**). All these results have demonstrated the success of coupling RoAda and MSTI,  as well as the well-obtained goal of our work.  We will also incorporate these discussion and new results into our revised manuscript.
>
> **W5. Justify the data summarization module and potential improvement.** For this module, we are aimed to extract a snapshot of the data that captures the representative characteristics and typical pattern summarization of the data, so  that it can play the role of task prompt. We utilized a straightforward yet effective MLP with an activation function to capture data snapshots, efficiently reflecting the intrinsic properties of the data. This approach has been empirically validated through ablation studies, demonstrating its capability to capture essential data features effectively. Thanks for your suggestion, and it inspire us for a potential improvement, i.e., we can further introduce contrastive loss, and build a constraint mechanism that is similar among tasks and different across tasks for the abstract representations of different tasks, so as to generate a higher quality task description learner. We are committed to further exploring this improvement and will include the results in revisions of our manuscript.
>
> Thank you once again for your thoughtful feedback. We are committed to addressing these issues thoroughly and improving our manuscript accordingly, ensuring it meets the standards expected by the NeurIPS community.
>
> Authors of Paper 2077

---

> > ### Comment · Reviewer_xkVJ · 2024-08-07
> >
> > Thank you for your response. My concerns are addressed. And I have raise my score.

---

> > > ### Author Response · Authors · 2024-08-07
> > > **Thanks for your positive feedback**
> > >
> > > Dear Reviewer xkVJ,
> > >
> > > We sincerely appreciate your constructive feedback and valuable comments, which really contributed to the enhancement of our manuscript. We will try our best to improve the  quality of manuscript and make both codes and datasets open-sourced. Thanks very much!
> > >
> > > Authors of Paper 2077

---

### Official Review · Reviewer_GaXf · 2024-07-03

**Soundness:** 3
**Presentation:** 2
**Contribution:** 3
**Rating:** 6
**Confidence:** 4

**Summary:**

This paper proposes a Continuous Multi-task Spatio-Temporal Learning Framework CMuST to facilitate the task-level cooperation in spatiotemporal predictions (mainly for traffic related tasks). The model is composed of three components . Data representation and integration module processes and standardizes diverse urban data into a harmonized format. MSTI modules reduce the complex interactions within spatiotemporal data. RoAda modules iteratively captures the task-wise consistency and task-specific diversity.

This study is generally fine with a relatively novel method for spatiotemporal multi-task learning through prompting (although prompting studies in this field is being rapidly developed). The main contribution is how the prompting is handled. The main problem I found is that the text is not that easy to follow with many jargons, coined (complicated) phrases, e.g., continuous multi-task spatiotemporal learning is a bit confusing -  is it something related to continual (lifelong) learning?

**Strengths:**

S1: The paper provides open access to data and code.
S2: The paper introduces the first continuous multi-task spatio-temporal learning framework for joint modeling of tasks within the same spatio-temporal domain, which is generally a novel method.
S3: The paper validates the proposed model on multiple datasets and tasks, demonstrating its generalization ability.

**Weaknesses:**

W1: During the model construction process, the paper does not clearly address the inconsistency issue of feature C across different task data. It is recommended to provide detailed explanations either in the data preprocessing stage or within the "Data representation and integration module" of the model.
W2: For continuous task rolling, specific operational details of each task model from training to convergence (such as the number of epochs) are not mentioned in the paper.
W3: Typos such as "Compressedd" (line 235).
W4: Modify proprietary terms that lack detailed explanations, such as "PECPM" on line 124.

Overall, the paper falls short in its presentation. Hope that it could be revised to ease the understanding.

**Questions:**

Q1: The definition of "domain" in the paper appears somewhat ambiguous. Please clarify whether "domain" refers to different types of tasks (such as "pick up" vs. "drop off") or different geographical regions (such as "NYC" vs. "SIP").
Q2: Please explain the distinction between the symbols "H[. . . ,slice(s)]" and "Es". Currently, these two parts appear to belong to the same content.
Q3. For different tasks in the same city, are the input features the same?

**Limitations:**

Yes

---

> ### Author Rebuttal · Authors · 2024-08-06
>
> Dear Reviewer GaXf,
>
> Thank you for your thoughtful review and acknowledging the potential and contribution of our work. We appreciate your insightful comments, which have provided us with an opportunity to refine our manuscript and address critical aspects that will enhance the clarity and impact of our research.
>
> **W1. Feature inconsistency across Tasks.** Actually, the tasks we collected have consistent features ($C=3$: numerical value, time of day, day of week).  The investigated space and interval units are standardized, and context factors are mapped into same dimension with an MLP to aviod the diverse dimensions of raw contexts. The detailed data preprocessing process can be referred to the **Common issue 4** of the global response. On the other hand, if another task with inconsistent features, we will use task-specific MLP for observation encoding and a task-specific prediction head to transform features into consistent spatiotemporal interaction (MSTI), which is corresponding to unique prompt for each task. This approach ensures the unique features of each task to be handled  uniformly and effectively, thus the shared spatial and temporal encoders can enhance spatiotemporal representation during rolling training.
>
> **W2.** **Task rolling details.** For the rolling training of each task, we set a maximum of 100 epochs, as this number is based on observations from our training logs, where most tasks typically converge around 90 epochs. Additionally, we maintain a learning rate of 1e-5 during rolling training to prevent catastrophic forgetting. We will include these specific operational details in the revised manuscript to provide a clearer understanding of this process.
>
> **W3 & W4. Presentation issue.** Appreciate your careful reading, we will check thoroughly and correct the typos in the whole paper and modify the proprietary terms with detailed explanation.
>
> Specifically, PECPM refers to Pattern Expansion and Consolidation on Evolving Graphs for Continual Traffic Prediction [1] proposed in SIGKDD 2023.
>
> [1] Pattern Expansion and Consolidation on Evolving Graphs for Continual Traffic Prediction, SIGKDD, 2023
>
> **Q1. Definition of domain.** Actually, in our study, various domains correspond to different urban elements collected with different manners in a given city. For instance, in an urban system, it includes diverse elements such as taxi demands, crowd flow, traffic speed and accidents. We collect and organize various urban elements in a city into one integrated dataset. The goal of our work is to explore the integrated intelligence from various domains and enhance learning of each individual urban element. To this end, the concept of multi-task here is to forecast various elements from different domains in an integrated model.
>
> **Q2. Explanation of formalization.** Initially, $E_s$ represents the spatial embeddings at the beginning of the model's operations, and $H[..., slice(s)]$ refers to the spatial segment within the tensor $H$ that initially matches $E_s$. As the model processes, $H'[..., slice(s)]$ evolves to reflect these updates, making it distinct from the initial embeddings $E_s$ , so we adopt a uniform slice representation. We will clarify this issue and the dynamic nature of these embeddings in our revised manuscript.
>
> **Q3. Input features across tasks.** For different tasks within the same city, the input features can vary specific to each task, though they share the same dimensionality $C$. Currently, the tasks we have collected features with an input dimension $C=3$, which includes numerical values, time of day, and day of week as answered in **W1**. In our future work, we plan to collect more diverse datasets and conduct further experiments with various types of input features with different dimensions $C$ to accommodate broader ranges of urban tasks.
>
> We again show our great appreciation of your valuable efforts on our work. We will comprehensively take you and all other reviewers comments into consideration and try our best to polish our manuscript for satisfying the high-level requirement of NeurIPS community.
>
> Authors of Paper 2077

---

> > ### Comment · Reviewer_GaXf · 2024-08-07
> > **Response**
> >
> > Thank you for your response. I have raised my assessment.

---

> > > ### Author Response · Authors · 2024-08-07
> > > **Thanks for your support and thoughtful comments**
> > >
> > > Dear Reviewer GaXf,
> > >
> > > We sincerely thanks for your support and valuable comments of our work, which have played a great significant role in the quality of our manuscript. We will make our best efforts to improve the presentation of the manuscript and add corresponding experimental details, as well as make the code and dataset open source for reference and availability. We would appreciate it if you could give us your kind support during the discussion phase. Thank you very much.
> > >
> > > Authors of Paper 2077

---

### Official Review · Reviewer_SHU9 · 2024-07-08

**Soundness:** 4
**Presentation:** 3
**Contribution:** 4
**Rating:** 7
**Confidence:** 4

**Summary:**

This paper proposes a Continuous Multi-task Spatio-Temporal learning framework (CMuST) to enhance urban intelligence. CMuST introduces a Multi-dimensional Spatio-Temporal Interaction network (MSTI) for capturing complex data interactions and a Rolling Adaptation training scheme (RoAda) to iteratively update the model, simultaneously maintaining task uniqueness and leveraging shared patterns across tasks. The framework is validated through extensive experiments on datasets from three cities, demonstrating superior performance against existing methods.

**Strengths:**

S1. Well presentation. This paper is well-presented and well-organized, providing a clear and comprehensive overview of the proposed methods and their implications.

S2. Good significance. The  proposed CMuST can jointly model different tasks of spatiotemporal forecasting  within the same spatiotemporal domain. This approach not only reinforces individual correlated tasks from a collective perspective, but also helps  understand the cooperative mechanism within the dynamic spatiotemporal system.

S3. Sufficient and qualified technical contribution. The contribution and innovation of MSTI network lies in that effectively dissecting interactions across multiple data dimensions for improved spatiotemporal representation and commonality extraction, and RoAda training scheme  for ensuring model  to adapt to new tasks and continuously learn commonality and personalized patterns. The coupling of these two major components can well contribute to the ST learning field.

S4. New benchmark construction and good experiment designs. The construction of benchmark datasets for three cities enriches the research field and provides a solid foundation for evaluating the framework performance. Extensive experiment designs including robustness in data-scarce scenarios, visualized attention scores, and performance variation with task increasing, demonstrate the framework's superiority in enhancing individual tasks with limited data and providing insights into task-wise continuous learning.

**Weaknesses:**

1. In Section 4.4, there is missing detailed description on how to avoid catastrophic forgetting during task rolling adaptation. It would be beneficial if the authors could provide more experimental details in this regard.

2. Lacking comparison baselines. More baselines which are argued for unified spatiotemporal/time series learning, such as UniST, UniTime should be added for comparisons.

**Questions:**

1. In your CMuST, how can you avoid catastrophic forgetting during task rolling adaptation?

2. Do more comparison experiments with SOTA baselines will be better.

**Limitations:**

The limitations are discussed.

---

> ### Author Rebuttal · Authors · 2024-08-06
>
> Dear Reviewer SHU9,
>
> Thank you for your valuable feedback and for recognizing the contributions of our work. Your insights are greatly appreciated and will help us further improve the quality of our manuscript.
>
> **W1&Q1. Avoiding catastrophic forgetting.** Actually, to avoid catastrophic forgetting, we implement several strategies during the Rolling Adaptation (RoAda) phase. Firstly, we set the learning rate for each task to 1e-5. This helps to retain more knowledge from previous tasks and prevents the model from over-adjusting to new tasks. By maintaining a low learning rate, the model can incrementally learn new information while preserving the stability of previously learned tasks. Additionally, we use task-specific weights for each task, such as task prompts. This method allows us to absorb the common features across all tasks while independently preserving and updating task-specific parameters. This approach ensures that when learning new tasks, the model does not forget the knowledge gained from previous tasks, thereby preventing catastrophic forgetting and avoiding overfitting to new tasks.
>
> **W2&Q2. More comparison baselines.** We appreciate your feedback and agree that comparing against unified spatiotemporal/time series learning (such as UniST [1] and UniTime [2]) will better showcase the generalization capabilities of our model for multi-task learning within same urban system. In response to your suggestion, we have conducted additional experiments comparing our CMuST model with UniST and UniTime. The results of these comparisons are as follows:
>
> | **Model/Dataset** | **①/Ⅰ** | **①/Ⅱ** | **①/Ⅲ** | **①/Ⅳ** | **②/Ⅰ** | **②/Ⅱ** | **③/Ⅰ** | **③/Ⅱ** | **③/Ⅲ** |
> | ----------------- | ------- | ------- | ------- | ------- | ------- | ------- | ------- | ------- | ------- |
> | UniST/MAE         | 11.3865 | 13.0762 | 6.8942  | 5.8804  | 11.7461 | 0.6985  | 2.3551  | 2.0134  | 1.1186  |
> | UniST/MAPE        | 0.4610  | 0.4478  | 0.3261  | 0.3490  | 0.2465  | 0.2661  | 0.3916  | 0.4162  | 0.2508  |
> | UniTime/MAE       | 12.2874 | 14.9120 | 7.4723  | 6.4641  | 13.9172 | 0.6993  | 2.4564  | 2.0341  | 1.1292  |
> | UniTime/MAPE      | 0.4721  | 0.4760  | 0.3671  | 0.3719  | 0.2965  | 0.2713  | 0.3987  | 0.4254  | 0.2511  |
>
> (The symbols in the table are explained in the attached **PDF** of the global response.)
>
> The results of these additional comparisons will be incorporated into the next version of our manuscript, providing a thorough evaluation and demonstrating the practical benefits and advancements introduced by our approach.
>
> Thank you once again for your valuable feedback. Your suggestions and comments will significantly enhance the quality of our manuscript, and we are committed to making the necessary improvements to meet the high standards of the NeurIPS community.
>
> [1] UniST: A Prompt-Empowered Universal Model for Urban Spatio-Temporal Prediction, SIGKDD, 2024
>
> [2] UniTime: A Language-Empowered Unified Model for Cross-Domain Time Series Forecasting, WWW, 2024
>
> Authors of Paper 2077

---

> > ### Comment · Reviewer_SHU9 · 2024-08-08
> >
> > Thanks for the response. I have checked all the content and also my concerns are well-addressed. This work  did address a new problem and contributed new techniques in the field of spatiotemporal learning. I would like to raise my score to 7.

---

> > > ### Author Response · Authors · 2024-08-08
> > > **Thanks for your recognition and insightful suggestions**
> > >
> > > Dear Reviewer SHU9,
> > >
> > > Thank you for your recognition and valuable suggestions, we will carefully revise our manuscript, including adding more experimental details and baselines to further improve the quality for satisfying the high-level requirement of NeurIPS community. Thanks a lot!
> > >
> > > Authors of Paper 2077

---

### Official Review · Reviewer_YiFX · 2024-07-13

**Soundness:** 3
**Presentation:** 2
**Contribution:** 3
**Rating:** 5
**Confidence:** 4

**Summary:**

This work proposes a multi-task spatiotemporal learning framework that helps the model understand the relationships between multiple tasks. The specific contributions lie in proposing MSTI to model the multidimensional spatiotemporal data and RoAda to capture the commonality and personalization among multiple tasks.

**Strengths:**

1.	The author attempts to construct a spatiotemporal model for continuous multi-tasks, which is an attractive motivation with development potential.
2.	RoAda provides an executable technical solution for spatiotemporal continuous learning.

**Weaknesses:**

1.	The author mentions multi-task and multi-domain problems several times in the introduction, but these concepts are not intuitively introduced in the paper. Multi-task and multi-domain do not always have a unified consensus in the ST community. For example, does multi-task include regression tasks and classification tasks? Does multi-domain refer to different cities or different modes of transportation? The author should provide more specific scopes for these terms in the introduction.
2.	The author's method of handling ST dependencies is not novel. Using cross-attention techniques to model from the perspectives of temporal dimension, spatial dimension, and spatiotemporal relationships separately is a common processing paradigm in the ST community. The contribution of MSTI can be considered overstated.
3.	The research on ST prediction and continual learning in the related work section is insufficient, lacking analysis of advanced ST prediction models and continual learning models in recent years.
4.	The author argues that continual learning can help spatiotemporal models enhance generalization ability. The theory and experiments in the paper are insufficient to support this argument.
5.	The author neglects experiments on cold start problems.
6.	The comparative experiments did not achieve the best results on all tasks; reasons for this should be analyzed.

Overall, I believe this work proposes a very attractive challenge, but in the end with the old problem, it only solves a standard problem, i.e. the ST multi-task problem. The author proposes Rolling Adaptation to solve this problem, which is a contribution that cannot be ignored. However, this work is incomplete for many issues mentioned in the introduction are not addressed, the definition of tasks is confusing at the beginning, and there is a lack of a pseudocode algorithm to help readers understand the proposed method more accurately.

**Questions:**

1.	Is the author using "continuous learning" to replace the commonly used "continual learning" in the community to indicate the uniqueness of this work?
2.	Did the author train three models on three datasets, or train only one model and continuously update it on three datasets? In other words, the author provided definitions for temporal increment, spatial increment, and feature increment in Definition 3, but in the actual work, was the increase in spatial nodes ignored?

**Limitations:**

The limitations have been addressed.

---

> ### Author Rebuttal · Authors · 2024-08-06
>
> Dear Reviewer YiFX,
>
> Thank you for your detailed and insightful feedback. We have carefully addressed each concern below.
>
> **W1. Concept, definition and scope of Multi-task learning.** Various domains correspond to different urban elements in a given city, and the concept of multi-task is to forecast various urban elements in a same neural network. (More details in **Common issue 1)**.
>
> **W2. Contribution of MSTI.** Previous attention-based spatiotemporal learners often process spatial and temporal aspects respectively. Different from those, MSTI designs cross-dimension attention, allowing flexible decomposition and spatial-temporal cross interactions. (More details in **Common issue 2**)
>
> **W3. Related work review.** TrafficStream [1], PECPM [2] on ST learning, and CLS-ER [3] on task-level continuous learning are investigated in our paper.  We supplement some more related works as below.
>
> 1. *ST learning:* DG2RNN [4] designs a dual-graph convolutional module to capture local spatial dependencies from both road distance and adaptive correlation perspectives. PDFormer [5] designs a spatial self-attention and introduces two graph masking matrices to highlight the spatial dependencies of short- and long-range views. TESTAM [6] uses time-enhanced ST attention by mixture-of-experts and modeling both static and dynamic graphs. These solutions focus on accuracy and generalization but neglect significance of continuous learning and fail to capture commonalities across different dynamic elements for collective intelligence.
> 2. *Detailed continuous learners:* C-LoRA [7] is low-rank adaptive by continuous self-regularization in cross-attention layer with stable diffusion. Kang et al. [8] develop a customized dirty label backdoor for online settings while maintaining high accuracy. COPAL [9] continuously adapts new data through a pruning approach guided by sensitivity analysis. These researches predominantly centers on the same task. Even CLS-ER addresses different tasks, it still focuses on image classification, trapping into little exploration on task-level streaming data. To model various aspects of commonality, MSTI and RoAda are proposed respectively. Therefore, the proposed task and solution are both novel to ST learning community.
>
> [1] TrafficStream, IJCAI'21
>
> [2] PECPM, KDD'23
>
> [3] Learning Fast, Learning Slow, ICLR'22
>
> [4] DG2RNN, TITS'24
>
> [5] PDFormer, AAAI'24
>
> [6] TESTAM, ICLR'24
>
> [7] C-LoRA, TMLR'24
>
> [8] Poisoning Generative Replay in Continual Learning ...., ICML'23
>
> [9] COPAL, ICML'24
>
> **W4. Generalization experiments and theory.**
>
> 1. *Experimental Evidence:* We have conducted generalization tests in Tab. 2 of Sec.5.2 and Fig. 4(b) of Sec. 5.4. Tab. 2 shows performance variations when node number reduced on NYC, indicating the robustness of CMuST. Fig. 4(b) shows  performance changing with number of input  tasks, indicating that task learning benefits from collective intelligence through assimilating common representations and interactive information, supporting the enhanced generalization in continual learning. Additional task-level cold start experiments are added to validate CMuST (**Common issue 3**).
> 2. *Theoretical Perspective:* It can be analyzed from uncertainty and information theory. First, introducing more diverse  samples and iteratively repeating model training can reduce the epistemic uncertainty and increase the experience of models [10,11]. From information-theory aspect, continual learning allows the model to maintain useful common information and dynamically updates the model with new data, increasing the mutual information across task-level observations [12,13]. By learning multiple related tasks, the perceived knowledge of model is expanded via increasing patterns and dependencies, leading to enhanced generalization [14,15].
>
> [10] Aleatoric and epistemic uncertainty in machine learning, MachLearn'21
>
> [11] SDE-Net, ICML'20
>
> [12] A Comprehensive Survey of Continual Learning, TPAMI'24
>
> [13] Graph information bottleneck, NeurIPS'20
>
> [14] Improving robustness to model inversion attacks via ..., AAAI'21
>
> [15] Incorporating neuro-inspired adaptability for continual learning ..., NMI'23
>
> **W5. Cold-start issue.** Generalization capacities have been empirically validated in Sec 5.2 and Sec. 5.4, where Tab. 2 can be viewed as imitating the cold-start issue on spatial dimension. (Task-level cold-start experiments are added in **Common issue 3**)
>
> **W6. Analysis of comparative experiments.**  1) CMuST achieves overall good performances with most best results, and only 4 out 18 achieve the second best, showing the superiority against all baselines. 2) Baseline models are usually designed from specific tasks and datasets, then they tend to be tailored and tuned for the specific data and tasks, e.g., PromptST is designed on NYC, thus obtaining best performances on NYC. To this end, baseline model tends to individually achieve best while CMuST achieves overall best results. We will incorporate such discussions into our manuscript.
>
> **Q1. Continuous & continual learning.**  'Continuous' is equivalent to 'continual'. The uniqueness of our work refers to a novel continuous task learning in ST community, which collects the integrated intelligence and benefits each individual learning task.
>
> **Q2. Model training and working details.** Given each dataset with different urban elements, we trained separate models for each city (dataset). Regarding the increment on spatial domain, we have conducted the generalization experiments on Tab.2 by node masking. The results suggest that CMuST can ease the data requirements of single task by capturing and exploiting commonalities and diversity among tasks.
>
> **Other. The pseudocode of RoAda.** We have added pseudocode of RoAda to global response **PDF**.
>
> Based on your suggestions, we are revising the manuscript to satisfy the high standards of the NeurIPS community. If you have further questions, please feel free to discuss with us.

---

> > ### Comment · Reviewer_YiFX · 2024-08-07
> >
> > Thank you for your response. I have raised the rating.

---

> > > ### Author Response · Authors · 2024-08-08
> > > **Thanks for your constructive suggestion and positive feedback**
> > >
> > > Dear Reviewer YiFX,
> > >
> > > We would like to express our deeply gratitude to your professional reviews and useful suggestions for promoting our manuscript. We promise to polish our manuscript, by improving the readability, supplementing more related works, and providing additional experiments. Many thanks! Hope you a nice day!
> > >
> > > Authors of Paper 2077

---

### Author Rebuttal · Authors · 2024-08-06

Dear Reviewers,

Thanks to all reviewers for your meticulous review and valuable feedback. We collated several common questions that identified multiple reviewers and  have compiled detailed explanations and responses to these concerns as follows:

**Common issue 1.(Reviewer YiFX, GaXf)** **The concept, definition and scope of Multi-task learning.** Actually, in our study, various domains correspond to different urban elements collected with different manners in a given city. For instance, in an integrated urban system, it includes taxi demands, crowd flow, traffic speed and accidents. We collect and organize various domain data (urban elements) in a city into one integrated dataset. The goal of our work is to explore the integrated intelligence from various domains and enhance learning of each individual urban element. To this end, the concept of multi-task here is to forecast various elements from different domains in an integrated model. Therefore, our work does not target at unifying regression or classification problems, but proposes an integrated model to iteratively establish the common intelligence among different elements and improve generalization for each element learning in succession, thus getting rid of task isolation. Noted that our experiments are performed with regression tasks, but it can easily generalize to classification task with shared representations.

**Common issue 2.(Reviewer YiFX, xkVJ) The design and technical contribution of MSTI.** Conventional attention-based spatiotemporal learners often process spatial and temporal aspects respectively [1-3]. Different from those, our MSTI designs a cross-dimension attention mechanism, where the dimension indicates the data representation on spatial aspect or temporal aspect. Our MSTI not only considers the self-correlation within spatial dimension, temporal dimension and main observations (e.g., taxi demands, flows), but also the interactions from main observation to spatial representation and main to temporal representation. This design allows flexible decomposition and capturing spatial-temporal cross interactions. Coupling with RoAda, we can flexibly capture the various commonality between spatial-temporal dimensions thus enhancing the continuous learning over each task. We believe this strategy is less-explored, especially for continuous multi-task ST learning.

[1] Learning Dynamics and Heterogeneity of Spatial-Temporal Graph Data for Traffic Forecasting, TKDE'22

[2] STAEformer, CIKM' 23

[3] PDFormer, AAAI'24

**Common issue 3.(Reviewer YiFX, xkVJ)** **Experiments for cold-start and generalization.** We have designed the experiment of cold start. Specifically, for NYC dataset, we selected three of the four tasks of Crowd In, Crowd Out, Taxi Pick and Taxi Drop in turn for training, and calculated the adaptation time and results for the remaining one task on this basis, comparing with training a single task alone. A similar design is applied for SIP and Chicago datasets. The results are shown in Table 2 in the attached **PDF**.

The results show that, both in terms of effect and time, it performs better than single task, indicating that our model adapts to the newly arrived task more quickly and well, which is conducive to solving the problem of cold start of urban prediction.

**Common issue 4.(Reviewer GaXf, xkVJ)** **Preparation of datasets and data processing.**

We construct datasets with multiple tasks for each city. For a given city, we first collect the intensive main ST data and filter out the in-range geolocation indicators (e.g., GPS). Different urban elements  are then aggregated to corresponding valid geographical ranges.
The space and interval units are standardized. The contexts are mapped into same dimension with an MLP to aviod the diverse dimension of raw contexts.

1. **NYC**: We collect yellow taxi trip data from January to March 2016 from the NYC Open Data website. Each trip record includes information such as pickup and dropout times, locations, and the number of passengers. We filter out records with abnormal longitude and latitude values or missing data. Then we select data within Manhattan and surrounding areas, divided into 30x15 grids, and counted trips per grid, selecting those with total trips greater than or equal to 1000, resulting in 206 grids. Each grid's data is aggregated into 30-minute intervals, yielding taxi pickup counts, taxi dropout counts, and crowd in/out flows. We also include time of day (tod) and day of week (dow) as context, resulting in four tasks with input features [value, tod, dow].
2. **SIP**: We collect traffic data from Suzhou Industrial Park from January to March 2017, comprising tens of thousands of records. The area is divided into nodes, and data is aggregated into 5-minute intervals. After filtering out grids with sparse data, we obtain 108 nodes, each containing traffic speed and traffic flow. We include time of day and day of week as input context, resulting in two tasks: traffic flow and traffic speed, with input [value, tod, dow].
3. **Chicago**: We collect taxi trip and accident data from the Chicago Open Data platform for June to December 2023. The taxi data includes trip start, end times and locations. We divide the area into 30x20 grids and select grids with total trips greater than 100, resulting in 220 grids. Similar to the NYC dataset, data is aggregated into 30-minute intervals, yielding taxi pickup and dropout counts, resulting in two tasks with input features [value, tod, dow]. The accident data includes incident locations, times, casualty numbers, and injury severity of each casualty. We then obtain the risk score by weighting it according to each casualty and injury, mapped it to the 220 grids, and aggregated the risk score over time intervals, resulting in a risk task with input features [risk score, tod, dow].

The detailed process can be found in the code implementation in the anonymous repository.

Authors of Paper 2077

---

### Author Response · Authors · 2024-08-12
**Thanks for your time on our manuscript**

Dear AC and reviewers,

We would like to express our great appreciation of your valuable time on our manuscript. As the deadline of reviewer-author discussion period is approaching, we wonder where there are any further questions on our work and we are always willing to resolve them, thus help better understanding of this work on a novel task-level continuous ST learning. We finally thank you all can support our work during the successive stages of discussions and we promise to public all the resources of this work including both datasets and codes if accepted.

Sincere thanks!

Authors of Paper  2077

---

### Decision · Program_Chairs · 2024-09-25

**Decision:**

Accept (oral)

**Comment:**

The paper presents a Continuous Multi-task Spatio-Temporal learning framework (CMuST) that introduces a Multi-dimensional Spatio-Temporal Interaction network (MSTI) for capturing complex data interactions and a Rolling Adaptation training scheme (RoAda) to iteratively update the model, simultaneously maintaining task uniqueness and leveraging shared patterns across tasks.The main goal of the work (as summarized by the authors in their rebuttal) is to explore the integrated intelligence from various domains and enhance learning of each individual urban element. To this end, the concept of multi-task here is to forecast various elements from different domains in an integrated model. The framework is validated on data from 3 cities to enhance urban intelligence.

The paper presents novel technical contributions as summarized by the various reviewers. The benchmark construction and experimental designs are also novel. Also, all reviewers have agreed that the authors have addressed their concerns.

As the paper presents a novel framework, with a novel continual learning method for spatio-temporal data sets aimed at enhancing urban intelligence, it will be of interest to a larger audience in the domain.